# Absolute quantification of translational regulation and burden using combined sequencing approaches

Thomas E Gorochowski[1,2,*] (iD), Irina Chelysheva[3], Mette Eriksen[4] (iD), Priyanka Nair[3], Steen Pedersen[4] & Zoya Ignatova[3,**] (iD)

## Abstract

Translation of mRNAs into proteins is a key cellular process. Ribosome binding sites and stop codons provide signals to initiate and terminate translation, while stable secondary mRNA structures can induce translational recoding events. Fluorescent proteins are commonly used to characterize such elements but require the modification of a part's natural context and allow only a few parameters to be monitored concurrently. Here, we combine Ribo-seq with quantitative RNA-seq to measure at nucleotide resolution and in absolute units the performance of elements controlling transcriptional and translational processes during protein synthesis. We simultaneously measure 779 translation initiation rates and 750 translation termination efficiencies across the *Escherichia coli* transcriptome, in addition to translational frameshifting induced at a stable RNA pseudoknot structure. By analyzing the transcriptional and translational response, we discover that sequestered ribosomes at the pseudoknot contribute to a $\sigma^{32}$-mediated stress response, codon-specific pausing, and a drop in translation initiation rates across the cell. Our work demonstrates the power of integrating global approaches toward a comprehensive and quantitative understanding of gene regulation and burden in living cells.

**Keywords** genetic circuits; Ribo-seq; RNA-seq; transcription; translation
**Subject Categories** Genome-Scale & Integrative Biology; Methods & Resources; Synthetic Biology & Biotechnology
**Mol Syst Biol. (2019) 15: e8719**

## Introduction

Gene expression is a multi-step process involving the transcription of DNA into messenger RNA (mRNA) and the translation of mRNAs into proteins. To fully understand how a cell functions and adapts to changing environments and adverse conditions (e.g., disease or chronic stress), quantitative methods to monitor these processes are required (Belliveau *et al*, 2018). Gene regulatory networks (also known as genetic circuits) control when and where these processes take place and underpin many important cellular phenotypes. Recently, there has been growing interest in building synthetic genetic circuits to understand the function of natural gene regulatory networks through precise perturbations and/or creating systems *de novo* (Smanski *et al*, 2016; Wang *et al*, 2016).

The construction of a genetic circuit requires the assembly of many DNA-encoded parts that control the initiation and termination of transcription and translation. A major challenge is predicting how a part will behave when assembled with many others (Cardinale *et al*, 2013). The sequences of surrounding parts (Poole *et al*, 2000), interactions with other circuit components or the host cell (Cardinale *et al*, 2013; Ceroni *et al*, 2015; Gyorgy *et al*, 2015; Gorochowski *et al*, 2016), and the general physiological state of the cell (Wohlgemuth *et al*, 2013; Gorochowski *et al*, 2014) can all alter a part's behavior. Although biophysical models have been refined to capture some contextual effects (Salis *et al*, 2009; Seo *et al*, 2013; Espah Borujeni *et al*, 2014), and new types of part created to insulate against these factors (Moon *et al*, 2012; Daniel *et al*, 2013; Mutalik *et al*, 2013; Siuti *et al*, 2013; Yang *et al*, 2014; Shendure *et al*, 2017), we have yet to reach a point where large and robust genetic circuits can be reliably built on our first attempt. A crucial step toward this goal will be to better understand how the many parts of large genetic circuits function in concert. However, approaches to simultaneously measure the performance of many parts within the context of a circuit are currently lacking.

Fluorescent proteins and probes are commonly used to characterize the function of genetic parts (Jones *et al*, 2014; Hecht *et al*, 2017) and debug the failure of genetic circuits (Nielsen *et al*, 2016). For circuits that use transcription rate (i.e., RNAP flux) as a common signal between components (Canton *et al*, 2008),

---

1 BrisSynBio, University of Bristol, Bristol, UK
2 School of Biological Sciences, University of Bristol, Bristol, UK
3 Biochemistry and Molecular Biology, Department of Chemistry, University of Hamburg, Hamburg, Germany
4 Biomolecular Sciences, Department of Biology, University of Copenhagen, Copenhagen, Denmark
*Corresponding author. Tel: +44 1173 941465; E-mail: thomas.gorochowski@bristol.ac.uk
**Corresponding author. Tel: +49 40 42838 2332; E-mail: zoya.ignatova@uni-hamburg.de

debugging plasmids containing a promoter responsive to the signal of interest have been used to drive expression of a fluorescent protein to track the propagation of signals and reveal the root cause of failures (Nielsen *et al*, 2016). Alternatively, any genes whose expression is controlled by the part of interest can be tagged with a fluorescent protein (Snapp, 2005). Such modifications allow for a readout of protein level but come at the cost of alterations to the circuit. This is problematic as there is no guarantee the fluorescent tag itself will not affect a part's function (Baens *et al*, 2006; Margolin, 2012).

The past decade has seen tremendous advances in sequencing technologies. This has resulted in continuously falling costs and a growing range of information that can be captured (Goodwin *et al*, 2016). Sequencing also offers several advantages over fluorescent probes for characterizing and debugging genetic parts and circuits. Firstly, it does not require any modification of the circuit DNA. Second, it provides a more direct measurement of the processes being controlled (e.g., monitoring transcription of specific RNAs), and third, it captures information regarding the host response and consequently their indirect effects on a part's function. Recently, RNA sequencing (RNA-seq) has been used to characterize every transcriptional component in a large logic circuit composed of 46 genetic parts (Gorochowski *et al*, 2017). While successful in demonstrating the ability to characterize genetic part function, observe internal transcriptional states, and find the root cause of circuit failures, the use of RNA-seq alone restricts the method to purely transcriptional elements and does not allow for quantification in physically meaningful units.

Here, we develop an approach that combines ribosome profiling (Ribo-seq) with quantitative RNA-seq that enables the high-throughput characterization of endogenous sequences and synthetic genetic parts controlling transcription and translation in absolute units. Ribo-seq provides position-specific information on translating ribosomes through sequencing of ribosome-protected fragments (RPFs; approximately 25–28 nt). This allows for genome-wide protein synthesis rates to be calculated with accuracy similar to quantitative proteomics (Li *et al*, 2014). By supplementing the sequencing data with other experimentally measured cell parameters, we generate transcription and translation profiles that capture the flux of both RNA polymerases (RNAPs) and ribosomes governing these processes. We apply our method to *Escherichia coli* and demonstrate how local changes in these profiles can be interpreted using mathematical models to infer the performance of three different types of genetic part in absolute units. Finally, we demonstrate how genome-wide shifts in transcription and translation can be used to dissect the burden that synthetic genetic constructs place on the host cell and the role that competition for shared cellular resources, such as ribosomes, plays.

# Results

## Generating transcription and translation profiles in absolute units

To enable quantification of both transcription and translation in absolute units, we modified the RNA-seq protocol and extended the Ribo-seq protocol with quantitative measurements of cellular properties (red elements in Fig 1A). For RNA-seq, we introduced a set of RNA spike-ins to our samples at known molar concentrations before the random alkaline fragmentation of the RNA (left panel, Fig 1A). The RNA spike-ins span a wide range of lengths (250–2,000 nt) and concentrations and share no homology with the transcriptome of the host cell (Appendix Fig S1). Using RNA spike-ins with known concentrations, the mapped reads can be converted to absolute molecule counts and then normalized by cell counts to give absolute transcript copy numbers per cell (Mortazavi *et al*, 2008; Bartholomäus *et al*, 2016) (Materials and Methods). The total number of transcripts per cell was ~8,200 which correlates well with earlier measurements of ~7,800 mRNA copies/per cell using a single spike-in (Bartholomäus *et al*, 2016). Similar overall copy numbers have been theoretically predicted (Bremer *et al*, 2003) and experimentally determined for another *E. coli* strain (Taniguchi *et al*, 2010). For Ribo-seq, we directly ligated adaptors to the extracted ribosome-protected fragments (RPFs) (Guo *et al*, 2010) to capture low-abundance transcripts (Del Campo *et al*, 2015). Sequencing was also complemented with additional measurements of cell growth rate, count, and protein mass (right panel, Fig 1A).

A previous method was further developed to generate transcription profiles that capture the number of RNAPs passing each nucleotide per unit time across the entire genome (i.e., the RNAP flux). This assumes that RNA levels within the cells have reached a steady-state (Gorochowski *et al*, 2017) and that all RNAs have a fixed degradation rate (0.0067/s) so that RNA-seq data, which captures a snapshot of relative abundances of RNAs, can be used to estimate relative RNA synthesis rates (Gorochowski *et al*, 2017). Because each RNA is synthesized by an RNAP, these values are equivalent to the relative RNAP flux. Since mRNA degradation rates can vary significantly across the transcriptome, we relaxed this assumption by incorporating experimentally measured transcript-specific degradation rates using previously published data (Chen *et al*, 2015). Finally, by using the known molar concentrations of the RNA spike-ins and their corresponding RNA-seq reads from our modified protocol (Appendix Fig S1), we are able to convert the transcription profiles into RNAP/s units. Existing mathematical models of promoters and terminators were then used to interpret changes in the transcription profiles and quantify the performance of these parts in absolute units.

To generate translation profiles that capture the ribosome flux per transcript, we first took each uniquely mapped RPF read from the Ribo-seq data and considering the architecture of a translating ribosome, estimated the central nucleotide of each codon in the ribosomal P site, i.e., the peptidyl-tRNA site (Materials and Methods; Appendix Fig S6) (Mohammad *et al*, 2016). By summing these positions for all reads at each nucleotide $x$, we computed the RPF coverage $N(x)$. If each ribosome translates at a relatively constant speed, then at a point in time the RPF coverage is proportional to the number of ribosomes at each nucleotide. This captures relative differences in ribosome flux, i.e., more heavily translated regions will have a larger number of ribosomes than lowly translated segments and so accrue a larger number of RPF reads in the Ribo-seq snapshot. However, the translation rate of individual codons can vary causing an enrichment in RPF reads at slowly translating codons (Woolstenhulme *et al*, 2015). Therefore, we divide $N(x)$ by the translation time of the codon (Fluitt *et al*, 2007) with a central nucleotide at position $x$ to give the weighted RPF coverage $W(x)$.

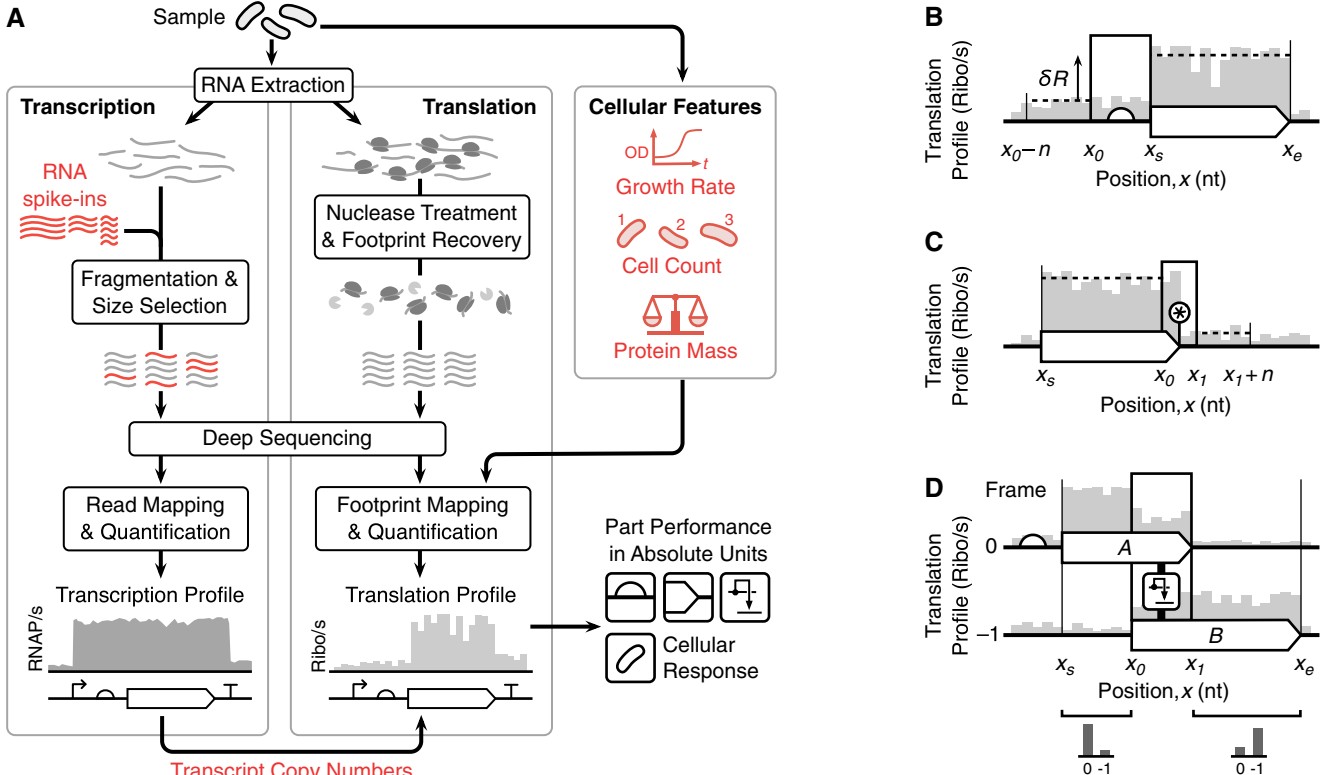

**Figure 1. Overview of the workflow.**

A　Major steps involved when quantifying transcription (RNA-seq) and translation (Ribo-seq) and the additional cellular features measured. Elements required for quantification in absolute units are highlighted in red.

B　Model for calculating the translation initiation rate of a ribosome binding site, see equation (2).

C　Model for calculating translation termination efficiency of a stop codon, see equation (3). Star denotes the location of the stop codon.

D　Model for calculating translational frameshifting efficiency between two coding regions "A" and "B" in zero and −1 reading frames, respectively, see equation (4).

This weighting corrects for position-specific variations. Moreover, the approach is extendable by other factors that may cause variations in translation speed, e.g., local mRNA secondary structure (Del Campo *et al*, 2015; Gorochowski *et al*, 2015) or interaction of some nascent chain segments with the ribosomal exit tunnel (Charneski & Hurst, 2013).

We next convert the weighted RPF coverage into a translation profile whose height corresponds directly to the ribosome flux across each nucleotide in ribosomes/s units. By assuming that each weighted RPF read corresponds to an actively translating ribosome which synthesizes a full-length protein product, and that the cellular proteome is at steady-state, then the protein copy number for gene $i$ is given by $n_i = \frac{f_i m_t}{f_t m_i}$. Here, $f_t$ is the weighted total number of mapped RPF reads, $m_t$ is the total protein mass per cell, and $f_i$ and $m_i$ are the weighted number of mapped RPF reads and the protein mass of gene $i$, respectively. We measured $m_t$ directly (Fig 1A) and calculated $m_i$ from the amino acid sequence of gene $i$ (Materials and Methods). Because proteins are synthesized by incorporating individual amino acids during the translocation cycle (i.e., by ribosome translocating from the A to P site), the replication of the entire proteome requires $r_t = \Sigma_i n_i a_i$ ribosome translocations, where $a_i$ is the number of amino acids in the protein encoded by gene $i$. Assuming that cells are growing at a constant rate with doubling time $t_d$, then the total ribosome

flux across the entire transcriptome per unit time is given by $q = 3r_t/t_d$. The factor of three accounts for ribosomes translocating at three-nucleotide registers (i.e., 1 codon/s = 3 nt/s). These calculations also assume that active protein degradation has a small contribution compared to dilution by cell division, which is reasonable in most cases. For example, > 93 % of the *Escherichia* coli proteome is not subject to rapid degradation with protein half-lives being well beyond cell doubling times during exponential growth and even starvation conditions (Nath & Koch, 1971).

Finally, the translation profile for nucleotide $x$ is calculated by multiplying the total ribosome flux $q$ by the fraction of active ribosomes $W(x)/f_t$ at that position and normalizing by the number of transcripts per cell of the gene being translated $m_x$, computed from the RNA-seq data (Fig 1A). This gives,

$$R(x) = \frac{qW(x)}{m_x f_t}. \tag{1}$$

Importantly, because both the transcription and translation profiles are given in absolute units (RNAP/s and ribosomes/s, respectively), they can be directly compared across samples without any further normalization.

## Characterizing genetic parts controlling translation

Genetic parts controlling translation alter ribosome flux along a transcript, and these changes are captured by the translation profiles. We developed new mathematical models to interpret these signals and quantify the performance of RBSs, stop codons, and translational recoding (e.g., ribosome frameshifting) in open reading frames (ORFs) at stable secondary structures.

In prokaryotes, RBSs facilitate translation initiation and cause a jump in the translation profile after the start codon of the associated gene due to an increase in ribosome flux originating at that location (Fig 1B). If initiation is rate limiting (Li *et al*, 2014), then the translation initiation rate of an RBS (in ribosomes/s units) is given by the increase in ribosome flux downstream of the RBS,

$$\delta R = \sum_{i=x_s}^{x_e} \frac{R(i) - C(i)}{(x_e - x_s)} - \sum_{i=x_0-n}^{x_0} \frac{R(i) - C^- - C^+}{n}. \tag{2}$$

Here, $x_0$ is the start point of the RBS, and $x_s$ and $x_e$ are the start and end points of the protein-coding region associated with the RBS, respectively (Fig 1B). By averaging the translation profile over the length of the protein-coding region, we are able to smooth out small localized fluctuations that might affect the measurement. A window of $n = 30$ nt (10 codons) was also used to average fluctuations in the translation profile upstream of the RBS; the averaging window is equal to the approximate length of a ribosome footprint. If the transcription start site (TSS) of the promoter expressing this transcript fell in the upstream window, then the start point $(x_0 - n)$ was adjusted to the TSS to ensure that the incoming ribosome flux is not underestimated. A similar change was made if the coding sequence was within an operon and the end of an upstream protein-coding region falls in this window. In this case, the start point was adjusted to 9 nt (3 codons) downstream of the stop codon of the overlapping protein-coding region. We also included correction factors to remove the effect of translating ribosomes upstream of the RBS that are not in the same reading frame as the RBS-controlled ORF and therefore may not fully traverse the coding sequence due to out-of-frame stop codons. These are given by,

$$C^- = \sum_{i=0}^{(x_0-n)/3} \frac{R(x_0 - n + 3i + 2)}{(x_0 - n)/3}, \tag{3}$$

$$C^+ = \sum_{i=0}^{(x_0-n)/3} \frac{R(x_0 - n + 3i + 1)}{(x_0 - n)/3}, \tag{4}$$

$$C(x) = \begin{cases} c^- + c^+, & x < s^- \wedge x < s^+ \\ c^-, & x < s^- \wedge x \geq s^+ \\ c^+, & x \geq s^- \wedge x < s^+ \\ 0, & \text{otherwise} \end{cases} \tag{5}$$

where $s^-$ and $s^+$ are the positions of the first out-of-frame stop codon downstream of $x_0 - n$ in the $-1$ and $+1$ reading frame, respectively. $C^-$ and $C^+$ capture the average out-of-frame ribosome flux in the region upstream of the RBS in the $-1$ and $+1$ reading frame, respectively, and $C(x)$ calculates the total sum of these ribosome fluxes that would reach nucleotide $x$ downstream of the RBS.

Ribosomes terminate translation and disassociate from a transcript when a stop codon (TAA, TAG or TGA) is encountered. This leads to a drop in the translation profile at these points (Fig 1C).

Although this process is typically efficient, there is a rare chance that some ribosomes may read through a stop codon and continue translating downstream (Arribere *et al*, 2016). Assuming that all ribosomes translating the protein-coding region are in-frame with the associated stop codon and do not frameshift prior to it, then the translation termination efficiency of the stop codon (i.e., the fraction of ribosomes terminating) is given by,

$$T_e = 1 - \frac{\sum_{i=x_1}^{x_1+n} R(i)/n}{\sum_{i=x_s}^{x_0} R(i)/(x_0 - x_s)}. \tag{6}$$

Here, $x_0$ and $x_1$ are the start and end nucleotides of the stop codon, respectively, $x_s$ is the start of the coding region associated with this stop codon, and $n = 30$ nt (codons) is the window, with the same width as described above, used to average fluctuations in the translation profile downstream of the stop codon (Fig 1C). If additional stop codons are present in the downstream window, the end point of this window $(x_1 + n)$ was adjusted to ensure that the translation termination efficiency of only the first stop codon was measured. A similar adjustment was made if the end of a transcript generated by an upstream promoter ends within this window.

Translation converts the information encoded in mRNA into protein whereby each triplet of nucleotides (a codon) is translated into a proteinogenic amino acid. Because of the three-nucleotide periodicity in the decoding, each nucleotide could be either in the first, second, or third position of a codon, thus defining three reading frames for every transcript. Consequently, a single mRNA sequence can encode three different proteins. Although synthetic biology rarely uses multiple reading frames, natural systems exploit this feature in many different ways (Tsuchihashi & Kornberg, 1990; Condron *et al*, 1991a; Giedroc & Cornish, 2009; Bordeau & Felden, 2014). In our workflow, the RPFs used to generate the translation profiles were aligned to the middle nucleotide of the codon residing in the ribosomal P site, providing the frame of translation. To characterize genetic parts that cause translational recoding through ribosomal frameshifting, we compared regions directly before and after the part. Strong frameshifting will cause the fraction of RPFs to shift from the original frame to a new one when comparing these regions with the frameshifting efficiency given by,

$$F_e = 1 - \frac{\sum_{i=x_1}^{x_e} R(i)/(x_e - x_1)}{\sum_{i=x_s}^{x_0} R(i)/(x_0 - x_s)}. \tag{7}$$

Here, $x_0$ is the nucleotide at the start of the region where frameshifting occurs, and $x_1$ is the end nucleotide of the stop codon for the first coding sequence (Fig 1D).

## Measuring genome-wide translation initiation and translation termination in *Escherichia coli*

We applied our approach to *E. coli* cells harboring a *lacZ* gene whose expression is induced using isopropyl β-D-1-thiogalactopyranoside (IPTG) (Fig 2A). After induction for 10 min, *lacZ* expression reached 14% of the total cellular protein mass (Appendix Table S1). Samples from non-induced and induced cells were subjected to the combined sequencing workflow (Fig 1A). Sequencing yielded between 41 and 199 million reads per sample (Appendix Table S2)

with no measurable bias across RNA lengths and concentrations (Appendix Fig S1), and a high correlation in endogenous gene expression between biological replicates ($R^2 > 0.96$; Appendix Fig S2). Distributions of mRNA copy numbers and RPF densities per gene were similar across conditions with RPF densities showing a broader spread than mRNA copy numbers (Appendix Fig S5).

Transcription and translation profiles were generated from these data and used to measure translation initiation rates of RBSs and translation termination efficiencies of stop codons across the genome. To remove the bias due to the RPF enrichment at the 5′-end of coding regions (Ingolia *et al*, 2009) (Fig 2B), $x_s$ was adjusted to 51 bp (17 codons) downstream of the start codon when estimating average ribosome flux across a coding region in Equations (2) and (6). To determine whether translation rates were fairly constant across each gene, we compared the number of RPFs mapping to the first and second half of each coding region. If the ribosomes traverse the coding sequence at a constant speed, then the two halves of a transcript should have a near identical RPF coverage. We found a high correlation between both halves for cells with non-induced and induced *lacZ* expression with less than ± 1.5-fold difference for 80% of all genes (Appendix Fig S3). This suggests a relatively constant speed of the ribosomes across each coding sequence but does not allow for comparisons between genes due to potential gene-wide biases, e.g., an enrichment in rare codons for a particular gene.

We characterized chromosomal RBSs in *E. coli* by assuming that each covered a region spanning 15 bp upstream of the start codon. Like background RPF levels, the correction factors in equation (5) applied during characterization of the RBSs were small, on average 0.06 and 0.1% of the ribosome flux through the coding region, both with and without inducing *lacZ* expression with IPTG, respectively. The translation initiation rates of the 779 RBSs we measured varied over two orders of magnitude with a median initiation rate of 0.18 ribosome/s (Fig 2C; Dataset EV1). This closely matches previously measured rates for single genes (Kennell & Riezman, 1977). A few RBSs of transcripts mostly related to stress response functions (*tabA*, *hdeA*, *uspA*, *uspG*), the ribosomal subunit protein L31 (*rpmE*), and some genes with unknown function (*ydiH*, *yjdM*, *yjfN*, *ybeD*) reached much higher rates of up to 3.4 ribosomes/s.

To estimate translation termination efficiency at stop codons, we analyzed regions that spanned 9 nt up and downstream of the stop codon (Fig 2B). We excluded overlapping genes and those bearing internal sites that promote frameshifting (Baggett *et al*, 2017), both of which break the assumptions of our model. In total, the translation termination efficiency of 750 stop codons was measured and their median translation termination efficiency across the transcriptome was found to be 0.974, with 249 of them (33% of all measured) having translation termination efficiencies > 0.99 (Fig 2D; Dataset EV2). Similar performance for both RBSs ($R^2 = 0.81$) and stop codons ($R^2 = 0.45$) was found between cells

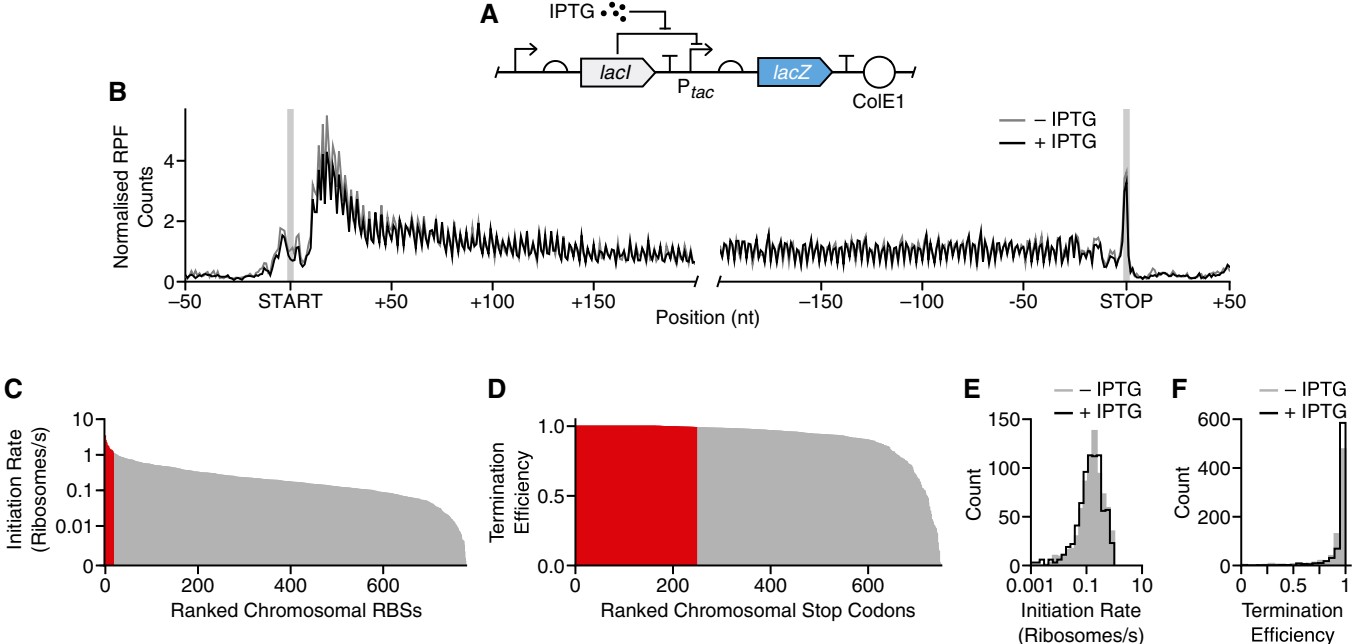

**Figure 2. Measuring translation initiation and translation termination signals across the *E. coli* transcriptome.**

A   Genetic design of the LacZ reporter construct whose expression is activated by the inducer IPTG.

B   Normalized RPF count profile averaged for all *E. coli* transcripts. Profiles generated for cells grown in the absence and presence of IPTG (1 mM). Start and stop codons are shaded.

C   Bar chart of all measured RBS initiation rates ranked by their strength. Strong RBSs with initiation rates > 1 ribosome/s are highlighted in red.

D   Bar chart of all measured translation termination efficiencies at stop codons ranked by their strength. Stop codons with translation termination efficiency > 0.99 are highlighted in red.

E   Distribution of initiation rates for cells grown in the absence and presence of IPTG (1 mM).

F   Distribution of translation termination efficiencies for cells grown in the absence and presence of IPTG (1 mM).

with non-induced and induced *lacZ* expression (Fig 2E and F; Datasets EV1 and EV2).

## Quantifying differences in transcription and translation of endogenous and synthetic genes

The quantitative measurements produced by our methodology allow both transcription and translation to be monitored simultaneously. To demonstrate this capability, we first focused on differences in the contributions of transcription and translation to overall protein synthesis rates of endogenous genes in *E. coli*. For each gene, we calculated the protein synthesis rate by multiplying the transcript copy number by the RBS-mediated translation initiation rate per transcript. We found a strong correlation with previously measured synthesis rates (Li *et al*, 2014) (Fig 3A). We also extracted the transcription and translation profiles of three genes (*uspA*, *ompA*, and *gapA*) whose protein synthesis rate was similar, but whose expression was controlled differently at the levels of transcription and translation (Fig 3B). Quantification of the promoters and RBSs for these three genes showed more than an order of magnitude difference in their transcription and translation initiation rates; *uspA* was weakly transcribed and highly translated, *ompA* was highly transcribed and weakly translated, and *gapA* was moderately transcribed and translated (Fig 3C).

Because we measure transcription and translation initiation rates in absolute units, it was also possible to determine their ratio (RNAP/ribosome) for each gene and assess whether there was a preference for high/low relative synthesis rates for transcription/translation given a gene's overall protein expression level. This analysis revealed a trend where weakly expressed genes exhibited low RNAP/ribosome ratios, while strongly expressed genes saw higher RNAP/ribosome ratios (Fig 3A).

These different modes of gene expression can have a major influence on the efficiency of protein synthesis (Ceroni *et al*, 2015) and affect the variability in protein levels between cells (Raser & O'Shea, 2005). For example, a metabolically efficient way to strongly express a protein of interest in bacteria is by producing high numbers of transcripts (e.g., with high transcription initiation rate and high stability) with a relatively weak RBS (e.g., low translation initiation rate). This ensures that each ribosome initiating on a transcript has a very low probability of colliding with others, guaranteeing efficient translation elongation (Cambray *et al*, 2018; Gorochowski & Ellis, 2018). We observe that this strategy is adopted for strongly expressed endogenous genes (Fig 3A).

We next sought to demonstrate the ability to measure dynamic changes in the function of regulatory parts using the LacZ construct. We quantified the inducible promoter and terminator controlling transcription, and the RBS and stop codon controlling translation when the inducer IPTG was absent and present. The transcription and translation profiles clearly showed the beginning and end of both the transcript and protein-coding region, with sharp increases and decreases at transcriptional/translational start and stop sites (Fig 3D). Induction caused a large increase in the number of *lacZ* transcripts from 0.18 to 110 copies per cell, which was directly observed in the transcription profiles. In contrast, the translation profiles remained stable across conditions. The $P_{tac}$ promoter has a transcription initiation rate of 0.0004 RNAP/s in the absence and 0.3 RNAP/s in the presence of IPTG (1 mM), respectively (Fig 3E). This

closely matches the previously measured transcription initiation rate of 0.33 RNAP/s for the $P_{lac}$ promoter (Kennell & Riezman, 1977), which the $P_{tac}$ promoter is derived from (De Boer *et al*, 1983). The RBS for the *lacZ* gene had consistent translation initiation rates of between 0.13 and 0.14 ribosomes/s, respectively (Fig 3E). It may seem counterintuitive to observe translation without IPTG induction because very few transcripts will be present. However, leaky expression from the $P_{tac}$ promoter was sufficient to capture enough RPFs during sequencing to generate a translation profile. It should be noted that the translation profile represents the ribosome flux per transcript; thus, its shape was nearly identical to that when the $P_{tac}$ promoter was induced. Like the RBS, both the transcriptional terminator and stop codon showed similar efficiencies of 0.93–0.95 and 0.96–0.99, respectively (Fig 3E).

## Characterizing a synthetic pseudoknot that induces translational recoding

Pseudoknots (PKs) are stable tertiary structures that regulate gene expression. They are frequently combined with slippery sequences in compact viral genomes to stimulate translational recoding and produce multiple protein products from a single gene (Tsuchihashi & Kornberg, 1990; Brierley *et al*, 2007; Giedroc & Cornish, 2009; Sharma *et al*, 2014). The percentage of recoding events generally reflects the stoichiometry of the translated proteins (e.g., capsule proteins for virus assembly) and helps overcome problems where the stochastic nature of transcription and translation makes maintenance of specific ratios difficult (Condron *et al*, 1991a). PKs are the most common type of structure used to facilitate mostly −1 frameshifting (Atkins *et al*, 2016) and in much rarer cases +1 frameshifting (e.g., in eukaryotic antizyme genes) (Ivanov *et al*, 2004). PKs consist of a hairpin with an additional loop that folds back to stabilize the hairpin via extra base pairing (Fig 4A). In addition to stimulating recoding events, PKs regulate translational initiation, where they interfere with an RBS through antisense sequences that base pair with the RBS (Unoson & Wagner, 2007; Bordeau & Felden, 2014). They also act as an evolutionary tool, reducing the length of sequence needed to encode multiple protein-coding regions and therefore act as a form of genome compression.

Two elements signal and stimulate frameshifting. The first is a slippery site consisting of a heptanucleotide sequence of the form XXXYYYZ which enables out-of-zero-frame paring in the A or P site of the ribosome, facilitating recoding events. The second is a PK situated 6–8 nt downstream of the slippery site. In bacteria, the distance between the slippery site and the 5′-end of the PK positions mRNA in the entry channel of the 30S ribosomal subunit, enabling contact with the PK which pauses translation and provides an extended time window for frameshifting to occur (Giedroc & Cornish, 2009).

To demonstrate our ability to characterize this process, we created an inducible genetic construct (referred to as PK-LacZ) that incorporated a virus-inspired PK structure within its natural context (*gene10*) fused to *lacZ* in a −1 frame (Fig 4A) (Tholstrup *et al*, 2012). *Gene10* ends with a stop codon such that translation of *lacZ* requires frameshifting at the PK. We specifically chose a PK variant (22/6a), which exhibits much lower frameshifting efficiency (~3%) (Tholstrup *et al*, 2012) compared to the wild-type PK (~10%) in its natural context (Condron *et al*, 1991a), but is known to heavily sequester and stall ribosomes and induce a significant stress

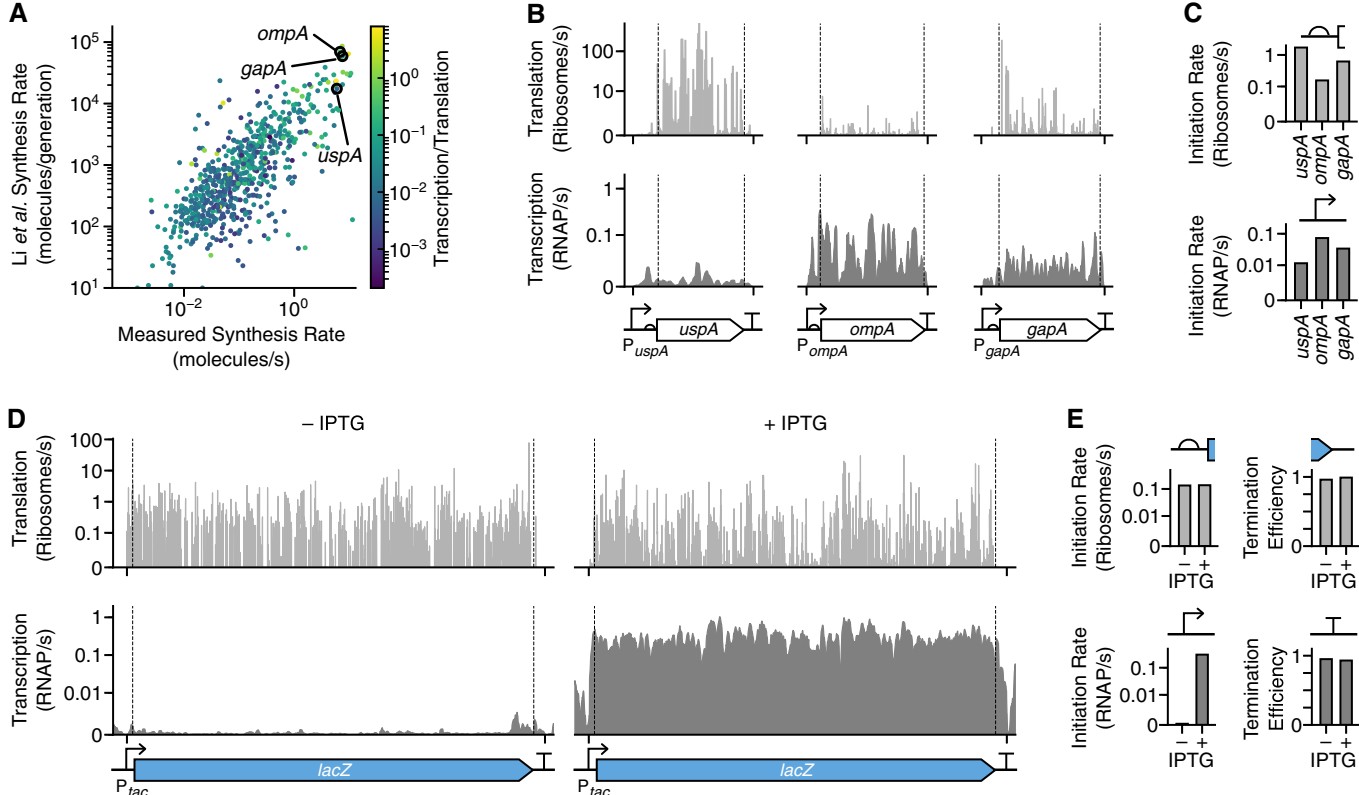

**Figure 3.  Simultaneous quantification of transcription and translation of endogenous genes and a synthetic genetic construct.**

A  Comparison of protein synthesis rate of endogenous *E. coli* genes measured using Ribo-seq from this study (in molecules/s units) and from that by Li *et al* (2014) (in molecules/generation units). Each point corresponds to a single gene, and color denotes the ratio of transcription initiation rate to translation initiation rate (giving RNAP/ribosome) capturing whether transcription (light yellow) or translation (dark blue) is more dominant.

B  Transcription (bottom) and translation (top) profiles for *uspA*, *ompA*, and *gapA*, computed from the RNA-seq and Ribo-seq data without induction. Positions of the genetic parts and gene are shown below the profiles.

C  Promoter strengths in RNAP/s units and RBS initiation rates in ribosome/s units.

D  Transcription (bottom) and translation (top) profiles for *lacZ*. Profiles are shown for cells in the absence and presence of IPTG (1 mM). Position of genetic parts and gene is shown below the profiles. RBS is omitted from the genetic design due to its size.

E  Measured promoter strength in RNAP/s units, RBS initiation rate in ribosomes/s units, and the transcriptional terminator and translation termination efficiency for *lacZ*. Data shown for cells in the absence and presence of IPTG (1 mM).

response (Tholstrup *et al*, 2012). With our approach, we sought to perform complementary quantification of the frameshifting efficiency, but more importantly to explore why such significant cellular stress was caused. A slippery site UUUAAAG preceded the PK. *Gene10* of bacteriophage T7 produces two proteins, one through translation in the zero-frame and one through a −1 frameshift; both protein products constitute the bacteriophage capsid (Condron *et al*, 1991b). We generated translation profiles to assess ribosome flux along the entire construct (Fig 4B). These showed high levels of translation up to the PK with a major drop of 80–90% at the PK to the end of the *gene10* coding region, and a further drop of ~97% after this region (Fig 4B). To analyze frameshifting within *gene10*, we divided the construct into three regions: (i) the *gene10* segment up to the slippery site, (ii) the middle region, which covers the slippery site along with the PK up to the *gene10* stop codon, and (iii) the downstream *lacZ* gene in a −1 frame. The large drops in the translation profiles at both the PK and *gene10* stop codons lead to low numbers of RPFs across the *lacZ* gene and caused high levels of noise in the translation profiles (Fig 4B). This made direct

comparisons of frame-specific expression levels at a codon resolution impossible. Therefore, for each of the three regions, we pooled the RPFs and calculated the fraction of RPFs in each frame as a total of all three possible frames. We found that the zero and −1 frames dominate the *gene10* and *lacZ* regions, respectively, with > 46% of all RPFs being found in these frames (top row, Fig 4C). The middle region saw a greater mix of all three, and the zero-frame further dropped in the *lacZ* region. This is likely due to a combination of ribosomes that have passed the PK successfully and terminated in zero-frame at the end of *gene10* and those that have frameshifted. Similar results were found with and without induction by IPTG (Fig 4C). An identical analysis of the reading frames from the RNA-seq data revealed that no specific frame was preferred with equal fractions of each (bottom row, Fig 4C). This suggests that the reading frames recovered for the RPFs were not influenced by any sequencing bias. We further tested whether the major translation frame could be recovered by analyzing the entire genome and measured the fraction of each frame across every gene. The correct zero-frame dominated in most cases (Fig 4D).

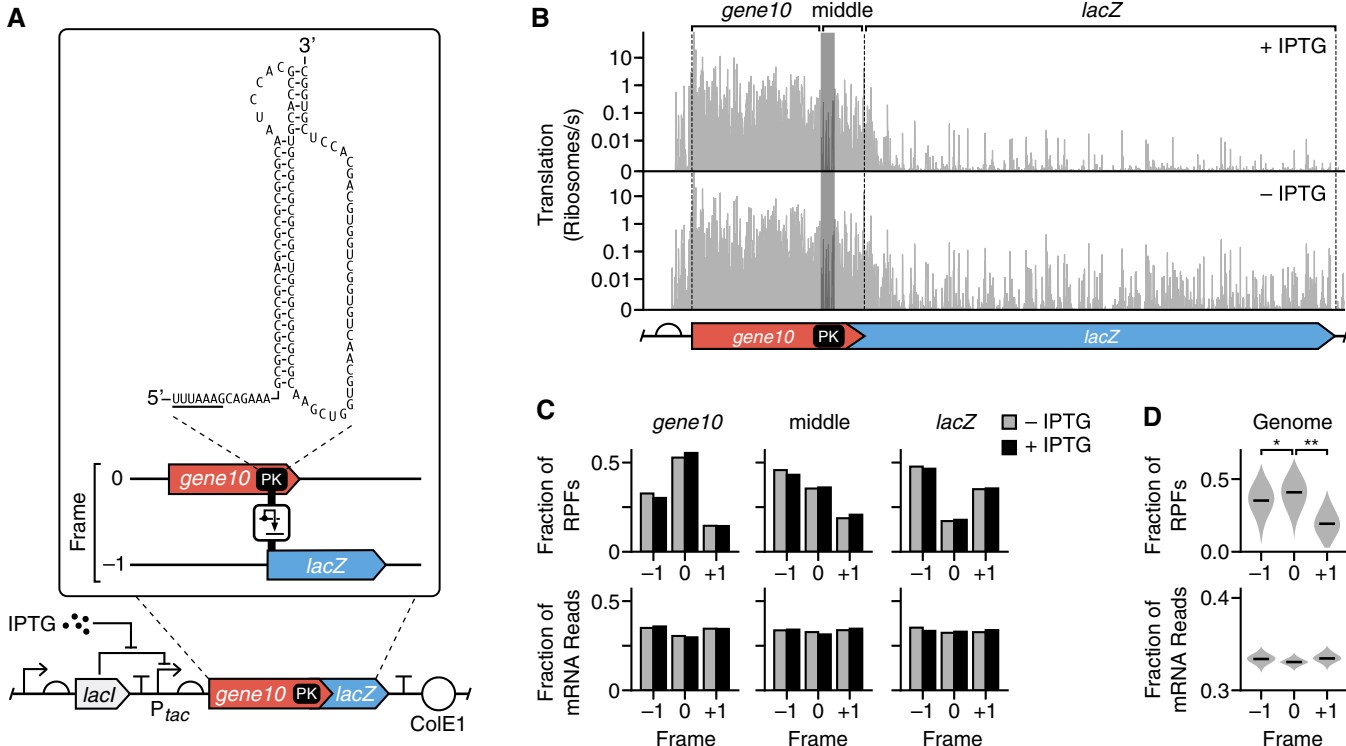

**Figure 4. Characterization of a synthetic pseudoknot construct that induces translational frameshifting.**

A  Genetic design of the PK-LacZ construct. Expanded sequence shows the PK secondary structure with the slippery site underlined, as well as the two genes (*gene10* and *lacZ*) in differing reading frames.

B  Translation profiles for the PK-LacZ construct in cells cultured in the absence (bottom) and presence (top) of IPTG (1 mM). The *gene10*, middle, and *lacZ* regions are labeled above the profiles. Shaded region denotes the PK, and dashed lines denote the start codon and stop codons of *gene10* and *LacZ*.

C  Fractions of the total RPFs and mRNA reads in each reading frame for the *gene10*, PK or middle, and *lacZ* regions. Data shown separately for cells cultured in the absence and presence of IPTG (1 mM).

D  Violin plots of the distributions of fractions of total RPFs and mRNA reads in each reading frame for all *E. coli* transcripts. Median values shown by horizontal bars. Data from two biological replicates. *$P$ = 0.049; **$P$ = $1.6 \times 10^{-9}$ (Mann–Whitney $U$ test).

Finally, to calculate the efficiency of PK-induced frameshifting, we compared the density of RPFs per nucleotide for the middle and *lacZ* regions. Because the PK causes ribosome stalling, the assumption of constant ribosome speed is broken for the *gene10* region upstream of the PK. Therefore, when calculating the frameshifting efficiency using equation (7), $x_s$ and $x_0$ were set to the start and end nucleotides of the middle region, directly downstream of the PK where pausing was not expected to occur. We found that the PK caused 2–3 % of ribosomes to frameshift. This precisely matched previous measurements of 3 % for the same PK variant (22/6a) measured by monitoring radioactive methionine incorporation (Tholstrup *et al*, 2012).

## Cellular response to a strong synthetic pseudoknot

Expression of strong PKs can severely impact cell growth, but the reason for this remains unclear (Tholstrup *et al*, 2012). We observed a large number of RPF reads within the *gene10* region (Fig 4B). These could be caused by either premature termination of ribosomes or stalled translation at the PK. Given previous experimental characterization of the 22/6a PK variant, which has been shown to sequester ribosomes (Tholstrup *et al*, 2012), it is likely

that many of these reads (Fig 4B) capture stalled ribosomes. Stalling increases the abundance of partially synthesized protein products but also limits the availability of translational resources, raising the question as to whether the expression of the PK-LacZ construct elicits cellular stress by sequestering ribosomes. To better understand the burden that expression of both *lacZ* and *PK-lacZ* exhibited on the cell, we compared shifts in transcription (i.e., mRNA counts) and translation efficiency (i.e., density of ribosome footprints per mRNA) of endogenous genes following induction with IPTG (Fig 5A; Dataset EV3). No major changes were observed for the LacZ construct (Fig 5A). In contrast, the PK-LacZ construct caused significant shifts in the expression of 491 genes (Dataset EV4). Of these, 341 were transcriptionally (i.e., significant changes in mRNA counts) and 204 translationally regulated (i.e., significant changes in translational efficiency), with little overlap (54 genes) between the two types of regulation (Fig 5B). Of the transcriptionally regulated genes, most saw a drop in mRNA counts, while translationally regulated genes were split between increasing and decreasing translational efficiencies. Gene ontology (GO) analysis revealed a clustering of transcriptionally downregulated genes in categories mostly linked to translation, e.g., ribosomal proteins, amino acid biosynthesis, amino acid activation (aminoacyl synthetases), and genes

involved in respiration and catabolism (Dataset EV5). Transcriptionally upregulated genes were associated with ATP binding, chaperones (*ftsH*, *lon*, *clpB*, *dnaJK*, *groLS*, *htpG*), ion binding, proteolytic activities (*ftsH*, *prlC*, *htpX*), and an endoribonuclease (*ybeY*). Interestingly, the expression of all of these are under $\sigma^{32}$ regulation which is the most common regulatory mode to counteract heat stress. $\sigma^{32}$ upregulation is often observed by expressing synthetic constructs, although the precise mechanism of $\sigma^{32}$ activation is not known (Ceroni *et al*, 2018). In our case, the incompletely synthesized polypeptides from the stalled ribosomes on the *PK-LacZ* mRNA are most likely partially folded or misfolded and generate misfolding stress similar to the heat shock response. Binding of the major *E. coli* chaperone systems, DnaK/DnaJ and GroEL/S, to the misfolded proteins negatively regulates $\sigma^{32}$. The shift of the chaperones to misfolded proteins releases $\sigma^{32}$, which then binds to the RNA polymerases and induces expression of heat shock genes (Guisbert *et al*, 2004; Mogk *et al*, 2011). This notion is supported by the fact that *dnaJ*, *groL/S*, and *grpE* were transcriptionally upregulated during PK induction as well as *ftsH*, which encodes the protease that degrades $\sigma^{32}$.

To test whether *PK-lacZ* expression caused changes in translation dynamics (e.g., ribosome pausing at particular codons), we next computed the occupancy of ribosomes at each codon (also known as codon occupancy) across the genome and compared it to that without inducing *PK-lacZ* expression (Appendix Fig S5) (Lareau *et al*, 2014). Notable increases in occupancy were found for the codons AGA, CTA, CCC, and TCC, which encode for arginine, leucine, proline, and serine, respectively (Fig 5C). All of these codons are rarely used in the genome for their cognate amino acid but were found in higher proportions across *gene10*. For example, the CTA codon that codes for leucine is only used by 4% of codons in the genome, while accounting for 8% of the *gene10* region. Coupled with the strong expression of *gene10*, the stress induced by this abnormal demand on cellular resources would be amplified.

The broad shifts in regulation at a cellular scale and changes in codon occupancy suggest that *PK-lacZ* expression may significantly limit the availability of shared cellular resources. From a translational perspective, this would manifest as a cell-wide drop in translation initiation rates as the pool of free ribosomes would be reduced (Gorochowski *et al*, 2016). To test this hypothesis, we compared the RBS initiation rates of endogenous genes before and after induction of *lacZ* and *PK-lacZ* expression and found a consistent reduction across all genes for both synthetic constructs (Fig 5D; Dataset EV1). While relatively small for the LacZ construct (5%) where no notable stress response was detected, the PK-LacZ construct triggered a large (54%) drop in translation initiation rates across the cell (Fig 5D). Analysis of the transcriptome composition and distribution of engaged ribosomes across cellular transcripts further revealed that the PK-LacZ construct made up 40% of all mRNAs and captured 47% of the shared ribosome pool engaged in translation (Fig 5E). This would account for the global drop in translation initiation rates and misfolding stress induced by the partially translated proteins from *gene10* transcripts, explaining the strong $\sigma^{32}$-mediated response.

We also observed a large difference in the number of transcripts for each construct after induction; the *lacZ* transcripts were 43-fold lower than those for *PK-lacZ* (81 versus 3,504 transcripts/cell, respectively). Such a difference is unlikely to occur solely through an increased transcription initiation rate at the $P_{tac}$ promoter.

Previous studies have shown that the decay rate of the *lacZ* transcript is highly dependent on the interplay between transcription and translation rates (Yarchuk *et al*, 1992; Iost & Dreyfus, 1995; Makarova *et al*, 1995). RNase E sites within the coding region become accessible to cleavage by RNase E when translation initiation rates are low because fewer translating ribosomes are present to sterically shield these sites and prevent degradation (Yarchuk *et al*, 1992). This mechanism could account for the lower *lacZ* transcript numbers, which in turn would reduce the number of sequestered ribosomes translating *lacZ* mRNAs and explain the lack of a stress response for this construct.

## Discussion

In this work, we present a new approach to quantify transcription and translation in living cells at a nucleotide resolution. This is based on a deep-sequencing workflow that combines Ribo-seq and quantitative RNA-seq with measures of key cellular parameters and uses mathematical models to interpret these data (Fig 1). We show that our high-throughput approach is able to simultaneously characterize the translation initiation rate of 779 RBSs and translation termination efficiency of 750 stop codons across the *E. coli* transcriptome (Fig 2), in addition to measuring the precise behavior of the genetic parts controlling transcription and translation of endogenous genes and a synthetic genetic construct that expresses *lacZ* (Fig 3). Because our methodology is based on sequencing, it can scale beyond the number of simultaneous measurements that are possible with common fluorescence-based approaches, and through the use of spike-in standards, we are able to indirectly infer part parameters in absolute units (i.e., transcription and translation rates in RNAP/s and ribosomes/s units, respectively). Furthermore, this work extends previous studies (Li *et al*, 2014; Owens *et al*, 2016; Baggett *et al*, 2017) by enabling the quantification of crucial expression parameters in absolute units and provides a more comprehensive and quantitative picture of both transcriptional and translational regulation across a cell.

To demonstrate the ability to quantitatively assess various translational processes that have been difficult to measure, we studied the behavior of a genetic construct that contains a strong virus-inspired PK structure that induces a translational frameshift (Fig 4). Following expression of *PK-lacZ*, the main reading frame shifts with a similar efficiency as measured previously for the same PK variant using radioactive methionine incorporation (Tholstrup *et al*, 2012). In contrast to *lacZ* expression, *PK-lacZ* also causes a major burden to the cell, sequestering a large proportion of the shared gene expression machinery, e.g., ribosomes (Fig 5). We observe transcriptome-wide increases in ribosome occupancy at codons rare for endogenous *E. coli* genes, but more frequent in the synthetic construct, suggesting that the strong expression of this gene places significant demands on the translational resources of the cell. This burden also results in significant changes in gene regulation (both transcriptional and translational), which was mediated by the alternative polymerase subunit, $\sigma^{32}$, that remodels the bacterial proteome following thermal stress (Guo & Gross, 2014). The likely cause of $\sigma^{32}$ activation is a combination of strong overexpression of *gene10* and misfolding stress triggered by partial unfolding of incompletely synthesized polypeptides (Giedroc & Cornish, 2009;

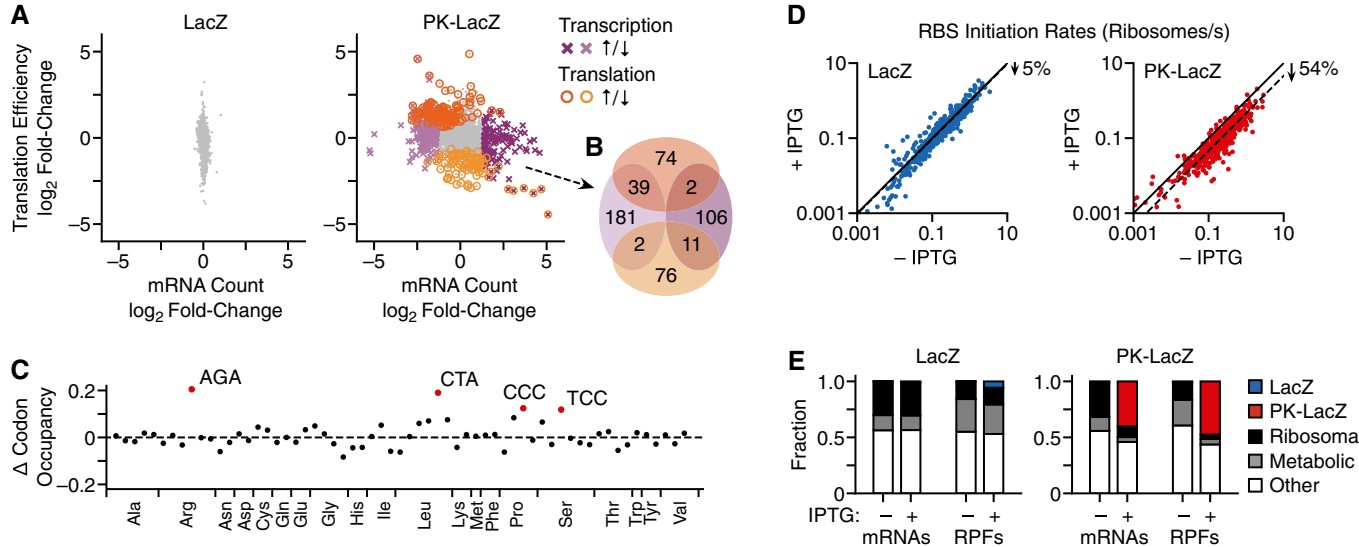

**Figure 5. Cellular response to the expression of a synthetic pseudoknot construct.**

A Change in expression of chromosomal genes in *E. coli* cells following induction of *PK-lacZ* expression (1 mM IPTG). Each point represents a transcript. Differentially expressed genes (mRNA count: *P* < 0.001 and absolute log$_2$ fold-change > 1.37; translation efficiency: *P* < 0.01) are highlighted in color and by an alternative point shape (transcriptional regulation: purple cross; translational regulation: orange open circle).

B Venn diagram of genes significantly regulated transcriptionally and translationally after induction of the PK-LacZ construct. Colors match those in panel (A).

C Change in codon occupancy for cells harboring the PK-LacZ construct after induction by IPTG (1 mM) calculated from the Ribo-seq data. Each point corresponds to a codon, which are ordered by amino acid identity and then by abundance in the genome (left most abundant, right least abundant). Dashed horizontal line denotes no change. Outliers are labeled and highlighted in red (Tukey test: 1.5 times the interquartile range below the first quartile or above the third quartile).

D Translation initiation rates for all *E. coli* RBSs in cells harboring the LacZ and PK-LacZ constructs in the absence and presence of IPTG (1 mM). Solid line shows the same initiation rate for both conditions. Dotted lines denote linear regressions for the data with no offset.

E Fractions of mRNA reads and RPFs mapping to each synthetic expression construct (LacZ and PK-LacZ) and *E. coli* transcripts, which are divided into three major categories: ribosomal, metabolic, and other functions. Data shown for cells cultured in the absence and presence of IPTG (1 mM).

Guo & Gross, 2014). To our knowledge, the stress response induced by a strong pseudoknot has not been reported before making this work a valuable data set for future studies.

Previous studies have used sequencing to investigate translational regulation. Ribo-seq was employed by Li *et al* (2014) to measure the protein synthesis rate of 3,041 genes and by Baggett *et al* (2017) to analyze translation termination at 1,200 stop codons. However, unlike our approach, which is calibrated by external RNA spike-in standards, these studies had no means of assessing the sensitivity of their measurements. Measuring the variability of several different RNA spike-in molecules at similar known molar concentrations allows us to accurately calculate a detection limit, emphasizing the benefit of including external standards in sequencing experiments. Furthermore, our ability to simultaneously capture in absolute units both mRNA copy numbers and protein synthesis rates enables us to move beyond changes at a single level (i.e., mRNA or protein), opening up new analyses focused on the multiple steps involved in protein synthesis (e.g., the number of ribosomes translating each mRNA). The use of absolute units in our work also provides several other benefits. First, it allows for biophysical constraints to be considered to help validate the feasibility of measurements and improve the quality of data (e.g., ensuring the rates measured do not exceed the maximum physically possible); second, it produces data that can be directly compared to any new experiment where measurements are also taken in absolute units. This includes using completely different organisms where the

composition and availability of gene expression machinery vary significantly. Such comparisons are impossible when using relative units (e.g., RPKMs) commonly employed in sequencing today which hinders data exchange, comparisons, and reuse.

A limitation of our approach is that the models underpinning the generation and interpretation of the transcription and translation profiles rely on some key assumptions that may not always hold true. For the transcription profiles to accurately capture RNAP flux, it is essential that the system has reached a steady-state because RNA-seq only measures RNA abundance at a single point in time and not directly the rate of RNA production (Gorochowski *et al*, 2017). While this assumption is valid for cells that have been exponentially dividing for several generations, rapidly changing RNA production or degradation rates (e.g., through increased expression of degradation machinery or a change in the growth phase) may cause issues. Furthermore, for quantification of absolute transcript numbers, while the RNA spike-ins will undergo the same depletion during sequencing library preparation, it is necessary to assume that the total RNA from the cells is efficiently extracted prior to this step. Incomplete cell lysis or low-efficiency RNA extraction would require a further correction during the quantification process.

For the translation profiles, the key assumptions are that every ribosome footprint gives rise to a full-length protein and that differences in codon translation times accurately capture localized changes in translation elongation speed along all transcripts. Translation is a complex multi-step process and can be affected by

ribosome pausing due to amino acid charge (Charneski & Hurst, 2013), premature translational termination (Freistroffer *et al*, 2000), and environmental conditions that alter cell physiology (Bartholomäus *et al*, 2016) or the global availability of cellular resources like ribosomes, tRNAs, and amino acids (Dong *et al*, 1996; Wohlgemuth *et al*, 2013; Gorochowski *et al*, 2016). Although these factors normally have only a small effect (Ingolia *et al*, 2009; Li *et al*, 2014; Del Campo *et al*, 2015), significant genome-wide shifts induced by long-term chronic stress can increase their occurrence and potentially alter translation elongation speed and processivity in a non-uniform way (Bartholomäus *et al*, 2016). Our calculation of absolute protein synthesis rates also relies on the assumption that proteins are stable with dilution by cell division dominating their degradation rate (Nath & Koch, 1971). This holds for most proteins, but care should be taken under severe stress conditions or for synthetic constructs where the proteome is heavily modified (e.g., by overexpressing proteases).

Being able to measure RNAP and ribosome flux across multi-component genetic circuits offers synthetic biologists a powerful tool for designing and testing new living systems (Nielsen *et al*, 2016; Gorochowski *et al*, 2017). These capabilities are particularly useful for large genetic circuits where many parts must function together to generate a required phenotype. Ideally, complex circuits are built by readily connecting simpler parts together. In electronics, this is made possible by using the flow of electrons as a common signal that captures the state at every point in a circuit. This signal can be easily routed between components using wires to create more complex functionalities. In genetic circuits, RNAP and ribosome fluxes can serve a similar role acting as common signals (Canton *et al*, 2008; Brophy & Voigt, 2014). Promoters and RBSs guide these signals to particular points in a circuit's DNA/RNA and allow them to propagate and be transformed.

The ability to easily connect large numbers of genetic parts allows for the implementation of complex functions, but can also lead to circuits that are fragile and break easily (Nielsen *et al*, 2016). This is particularly common for those that use components with sharp switch-like transitions (e.g., repressors with high cooperativity) (Nielsen *et al*, 2016). These types of part can lead to situations where although the output of the circuit behaves as desired, it becomes highly sensitive to changes in growth conditions or the inclusion of other genetic components (Gorochowski *et al*, 2017). This problem arises because the parts may be required to function near these sharp transition points allowing for minor perturbations to cause large deviations in expression that then propagates to the output of the circuit. The only way to guarantee the robustness of such systems is to either measure every internal state to ensure parts are not functioning near these transition points (Gorochowski *et al*, 2017), or to implement feedback control within the circuit itself for self-regulation (Ceroni *et al*, 2018). The ability to monitor every element in a circuit also makes our approach valuable when elucidating the root cause of failures. Instead of time-consuming tinkering with a circuit until the problem is found, our method allows for targeted modifications that precisely correct malfunctioning parts, accelerating developments in the field (Gorochowski *et al*, 2017).

Mature engineering fields rely on predictive models to efficiently develop complex systems by reducing the need to physically construct and test each design. To date, the accuracy of models in synthetic biology has been hampered by a lack of reliable, quantitative, and high-throughput measurements of genetic parts and devices, as well as their effects on the host cell. Attempts have been made to improve this situation by using standard calibrants to increase reproducibility across laboratories and equipment (Davidsohn *et al*, 2015; Beal *et al*, 2016; Castillo-Hair *et al*, 2016) and by including synthetic RNA spike-ins to enable absolute quantification of transcription (Owens *et al*, 2016). Our methodology complements these efforts by combining RNA-seq and Ribo-seq with RNA spike-in standards to quantify the regulation of transcription and translation. The importance of pushing biology toward measurements in absolute units has seen growing interest (Justman, 2018) and is becoming widely recognized as essential for developing mechanistic models that can support reliable predictive design (Endy *et al*, 2000; Jones *et al*, 2014; Belliveau *et al*, 2018). To demonstrate why, it is important to realize that many behaviors are intrinsically linked to their absolute scale. For example, the stochastic nature of biochemical reactions means that the inherent noise when only a few molecules are present will be far greater than when there are many. Therefore, knowing whether one arbitrary unit corresponds to 1 or 10,000 molecules is essential if the models are to hold any predictive power as to the expected variability. In this regard, the use of absolute measurements in mechanistic models of biological parts (Jones *et al*, 2014; Belliveau *et al*, 2018) and entire genetic systems (Endy *et al*, 2000) has already seen some success.

As we attempt to implement ever more complex functionalities in living cells (Nielsen *et al*, 2016) and push toward a deeper understanding of the processes sustaining life, scalable and comprehensive methodologies for quantitative measurement of fundamental processes become paramount. Such capabilities will move us beyond a surface-level view of living cells to one that allows the exploration of their innermost regulation and homeostasis.

# Materials and Methods

**Reagents and Tools table**

| Reagent/Resource | Reference or source | Identifier or catalog number |
|---|---|---|
| **Experimental models** | | |
| K12 (*E. coli*) | DSMZ | 498 |
| **Recombinant DNA** | | |
| pBR322 | Addgene | 31344 |

**Reagents and Tools table** (continued)

| Reagent/Resource | Reference or source | Identifier or catalog number |
|---|---|---|
| pBR322-LacZ | Tholstrup *et al* (2012) | N/A |
| pBR322-PK-LacZ | Tholstrup *et al* (2012) | N/A |
| **Oligonucleotides and other sequence-based reagents** | | |
| ERCC RNA Spike-In Mix | Ambion/Thermo Fisher Scientific | 4456740 |
| **Chemicals, enzymes, and other reagents** | | |
| RiboLock/SUPERaseIn | Thermo Fisher Scientific | EO0382 |
| MNase | New England Biolabs | M0247S |
| T4 Polynucleotide kinase | New England Biolabs | M0201S |
| T4 RNA ligase | New England Biolabs | M0202S |
| Pfu DNA Polymerase | Thermo Fisher Scientific | EP0502 |
| MICROBExpress mRNA isolation kit | Thermo Fisher Scientific | AM1905 |
| GeneJET RNA purification kit | Thermo Fisher Scientific | K0731 |
| µMACS Streptavidin kit | Miltenyi Biotec | 130-074-101 |
| Qubit dsDNA HS kit | Thermo Fisher Scientific | Q32851 |
| DNA1000 Chip | Agilent | 5067-1505 |
| **Software** | | |
| fastx-toolkit version 0.0.13.2 | https://github.com/agordon | |
| cutadapt version 1.8.3 | https://cutadapt.readthedocs.io/en/stable/ | |
| DESeq2 version 1.20 | https://bioconductor.org/packages/ | |
| GO.db version 2.1 | https://bioconductor.org/packages/ | |

## Strains, media, and inducers

The *E. coli* K12 strain, [K-12, *recA1 Δ(pro-lac) thi ara F':lacIq1 lacZ:: Tn5 proAB+*], harbors a pBR322-derived plasmid containing either *lacZ* with a fragment insert that contains a truncated *lac* operon with the $P_{tac}$ promoter and the wild-type *lacZ* under *lacI* control, or a pseudoknot-lacZ (*PK-lacZ*) consisting of *gene10*, a virus-derived RNA pseudoknot (Tholstrup *et al*, 2012), 22/6a, fused upstream of the *lacZ*. Bacteria were grown in MOPS minimal medium supplemented with 0.4% glycerol, 2.5 µg/ml vitamin B1, 100 µg/ml ampicillin, 20 µg/ml kanamycin, and additionally 50 µg/ml arginine for the *lacZ* expressing strain. The cells were grown for at least 10 generations at 37°C to ensure stable exponential growth before induction.

## Gene expression and preparation of the sequencing libraries

*LacZ* and *PK-lacZ* expression were induced with isopropyl β-D-1-thiogalactopyranoside (IPTG) to a final concentration of 1 mM at $OD_{600} \approx 0.4$ for 10 and 15 min, respectively. One aliquot of each culture was used to isolate RPFs and prepare the cDNA library for Ribo-seq as described in Bartholomäus *et al* (2016). In parallel, from another aliquot, total RNA was isolated with TRIzol (Invitrogen) and subjected to random alkaline fragmentation for RNA-seq as described in Bartholomäus *et al* (2016). Different than the previous protocol, prior to alkaline fragmentation, the total RNA was spiked in with RNA standards (ERCC RNA Spike-In Mix; Ambion) which were used to (i) determine the detection limit in each data set and (ii) calculate the copy numbers per cell. The RNA standards consist

of 92 different transcripts, covering lengths of 250–2,000 nt and approximately a 106-fold concentration range. Detection threshold (RPKM) has been set at values with a linear dependence between the reads from the spike-in controls and concentration in each RNA-seq data set. Spike-ins with linear correlation were used in the copy number analysis (Appendix Fig S1). Total protein concentration (grams of wet mass per ml culture) was determined by the Bradford assay using serial dilutions of the exponentially growing cells at different time points (e.g., prior the induction time at $OD_{600} = 0.4$ and following induction with 1 mM IPTG). Using the cell number and the volume of *E. coli* as 1 femtoliter, the protein mass was recalculated as grams of wet protein mass per cell.

## Processing of sequencing data

Sequenced reads were quality trimmed using fastx-toolkit version 0.0.13.2 (quality threshold: 20), sequencing adapters were cut using cutadapt version 1.8.3 (minimal overlap: 1 nt), and the reads were uniquely mapped to the genome of *E. coli* K-12 MG1655 strain using Bowtie version 1.1.2 allowing for a maximum of two mismatches. *LacZ* and other similar parts of the plasmids were masked in the genome. Reads aligning to more than one sequence including tRNA and rRNA were excluded from the data. The raw reads were used to generate gene read counts by counting the number of reads whose middle nucleotide (for reads with an even length, the nucleotide 5′ of the mid-position) fell in the coding sequence (CDS). Gene read counts were normalized by the length of the unique CDS per kilobase (RPKM units) and the total mapped reads per million (RPM units) (Mortazavi *et al*, 2008). Biological replicates were performed

for all sequencing reactions. Based on the high correlation between replicates (Appendix Fig S2), reads from both biological replicates were merged into metagene sets (Ingolia *et al*, 2009). Differential gene expression was performed using DESeq2 version 1.20. Firstly, transcripts with *P* < 0.01 for both translational efficiency and mRNA expression were selected. *P*-values were adjusted for multiple testing using false-discovery rate (FDR) according to Benjamini and Hochberg. Since the RNA-seq data sets have very high reproducibility between replicates (Appendix Fig S1), we decided to apply more restrictive threshold *P* < 0.001 and additionally selected the 25th percentile. The GO terms with significant enrichment (*P* < 0.01) were calculated using GO.db version 2.10.

### Calculating absolute transcript numbers

To calculate the transcript copy number, we used a method previously described by Bartholomäus *et al* (2016) and Mortazavi *et al* (2008). Briefly, the mapped reads for a transcript were related to the total reads and the length of the transcriptome. The latter was determined using the molecules of all spike-in standards above the detection limit (Appendix Fig S1) and was normalized by cell number.

### Calibration of ribosome profiling reads

RPFs were binned in groups of equal read length, and each group was aligned to the stop codons as described previously by Mohammad *et al* (2016). For each read length, we calculated the distance between the point a transcript leaves the ribosome and the middle nucleotide in the P site. This distance was used to determine the center of each P site codon along each mRNA (see Appendix Fig S6 for further details). As expected, the majority of our sequence reads were 23–28 nt and these read lengths were used for the further analysis. The ribosome occupancy per codon over the whole transcriptome was calculated as described by Lareau *et al* (2014), where the reads over each position within a gene were normalized to the average number of footprints across this gene. Metagene analysis of the ribosome occupancies within the start and stop codon regions was performed as described by Baggett *et al* (2017). Thereby, only genes with at least 5 RPFs in the chosen window were considered. Overlapping genes were excluded from the analysis.

### Data analysis and visualization

Data analysis was performed using custom scripts run with R version 3.4.4 and Python version 3.6.3. Plots were generated using matplotlib version 2.1.2, and genetic constructs were visualized using DNAplotlib version 1.0 (Der *et al*, 2017) with Synthetic Biology Open Language Visual (SBOLv) notation (Myers *et al*, 2017).

## Data availability

Sequencing data from RNA-seq and Ribo-seq are deposited in the Sequence Read Archive (https://www.ncbi.nlm.nih.gov/sra/) under accession number SRP144594.

**Expanded View** for this article is available online.

### Acknowledgements

We thank Alexander Bartholomäus for the initial mapping and earlier data analysis. This work was supported by BrisSynBio, a BBSRC/EPSRC Synthetic Biology Research Centre (grant BB/L01386X/1), a Royal Society University Research Fellowship (grant UF160357 to T.E.G.), the MOLPHYSX program of the University of Copenhagen (S.P.), and the European Union (grants NICHE ITN and SynCrop ETN to Z.I.)

### Author contributions

ZI and TEG conceived of the study. ME performed the sequencing experiments. PN performed the quantitative determination of cellular parameter. SP provided the LacZ and PK-LacZ constructs and advised the experimental acquisition of sequencing data. TEG developed the mathematical models. IC processed the sequencing data. ZI, TEG, and IC analyzed the data. ZI, TEG, and IC wrote the manuscript.

### Conflict of interest

The authors declare that they have no conflict of interest.

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
