## [Review Process File · Molecular Systems Biology]

Absolute quantification of translational regulation and burden using combined sequencing approaches

Thomas E. Gorochowski, Irina Chelysheva, Mette Eriksen, Priyanka Nair, Steen Pedersen and Zoya Ignatova.

Review timeline:

Submission date:	4 th November 2018
Editorial Decision:	6 th December 2018
Revision received:	7 th March 2019
Editorial Decision:	3 rd April 2019
Revision received:	9 th April 2019
Accepted:	15 th April 2019

Editor: Maria Polychronidou

Transaction Report:

1st Editorial Decision

6th December 2018

Thank you again for submitting your work to Molecular Systems Biology. We have now heard back from the three referees who agreed to evaluate your study. As you will see below, the reviewers raise a number of concerns, which unfortunately preclude the publication of the study in its current form.

The reviewers mention that as it stands the main conclusions are not well supported and point out that several assumptions could have a significant impact on the reported findings. However, considering that the reviewers appreciate that the addressed topic is important and the study seems likely to be useful for the field, we would like to offer you a chance to revise the study and address the points raised.

Without repeating all the comments listed below, the most fundamental issue that needs to be convincingly addressed is dealing with the effect of assumptions such as e.g. fixed RNA degradation rates and same translation elongation rates. The reviewers provide constructive suggestions in this regard. Another important issue refers to the need to clearly demonstrate that the proposed approach goes beyond what is possible to achieve using similar existing methodologies.

All other issues raised by the reviewers need to be satisfactorily addressed. As you may already know, our editorial policy allows in principle a single round of major revision so it is essential to provide responses to the reviewers' comments that are as complete as possible. Please feel free to contact me in case you would like to discuss in further detail any of the issues raised by the reviewers.

REFeree REPORTS

Reviewer #1:

This manuscript presents a quantitative analysis of mRNA abundance and translation in bacteria. Absolute mRNA abundances are estimated based on spike-in controls, whereas translation levels are calibrated based on steady-state protein levels. These calibrated data are used to describe the range of initiation and elongation rates for endogenous genes, as well as an inducible *lacZ* representative of heterologous protein expression. Having found that transcriptional induction of *lacZ* has little impact on its translation, the manuscript next analyzes a pseudoknot-dependent frameshift derived from bacteriophage. The frameshift is apparent in ribosome occupancy profiles of the transcript, and also induces broader physiological changes in cells that are attributed to sequestration of the translational machinery and the accumulation of unfolded or misfolded nascent proteins.

The absolute expression estimates presented in this manuscript are calibrated indirectly, depend on assumptions that are likely violated for many genes, and are not tested against any orthogonal measures. As such, the impact of these estimates over uncalibrated relative expression profiling are limited, and these fundamental caveats are not addressed. The cellular impact of pseudoknot overexpression is interesting, and may guide us to a deeper understanding of the fitness costs in heterologous protein expression. The current manuscript proposes interesting hypotheses about this effect, but does not test any of them.

My major concerns with the manuscript:

1. The calibration of absolute mRNA abundance based on spike-in controls is plausible, but these are then converted to absolute initiation rates with a very simplistic estimate that all mRNA decay rates are constant and equal. This is unrealistic, and limits the value of these estimates.
2. The calibration of absolute translation is even more limited. Absolute abundances are calibrated against total cellular protein content, dependent on the assumption that protein degradation is not a major factor in overall protein abundance. Again, this limits the value of the absolute estimates presented here.
3. No data is presented calibrating transcription or translation initiation rate estimates against any sort of true rate measurement (versus inference based on steady-state levels).
4. The manuscript presents an interesting analysis of relative transcription and translation rates (although limited as discussed above). However, these are interpreted as indicating genes that are "mostly governed by translation" versus those that are "mostly controlled by transcription". In the single, steady-state condition here, steady-state abundance is just the product of these two initiation rates and it isn't clear how to meaningfully partition "control". Control implies the extent to which changes in protein abundance result from changes in these synthesis rates (or decay rates, which are also regulated). At most this analysis can describe the range of initiation rates observed across different genes.
5. The manuscript reports that, " the PK caused 2-3% of ribosomes to frameshift, ~3-fold less than the 10% reported for the PK in its natural context". This should be verified by protein-level measurements - perhaps the quantitative estimate here (or in Condrón et al) is simply incorrect?
6. The manuscript reports, "...a large number of RPF reads within the gene10 region...many of these reads capture stalled ribosomes." What is the evidence or argument supporting this interpretation?

Reviewer #2:

The authors utilize RNA-Seq and Ribo-Seq together to measure mRNA levels and ribosome densities in engineered *E. coli* strains together with spike-in RNA controls. In one engineered strain, a synthetic promoter is used to inducibly express *lacZ*. In another engineered strain, a natural RNA pseudoknot within a viral gene10 is introduced upstream and out-of-frame of the *lacZ* coding sequence, creating the potential for translation frameshifting. By applying advanced data analysis,

the authors use their measurements to calculate the apparent translation initiation rates at start codons and the apparent translation termination rates at stop codons. They use these calculations together with spike-in RNA controls, total protein mass measurements, and per-protein mass calculations to estimate the absolute ribosome fluxes (ribosomes per second) across transcripts. By comparing RiboSeq and RNA-Seq measurements across their engineered strains, the authors identify significant changes in transcription and translation rates across many genes, likely due to the pseudoknot-mediated frameshifting and the resulting excess sequestration of ribosomes. Overall, the work presented is high-quality and interesting, demonstrating how RiboSeq and RNA-Seq can be used to quantify transcriptome and proteome-wide changes in cell physiology. However, the authors do their best to explain away some assumptions that could significantly alter their calculated results. The RiboSeq technique is fairly new and not without bias. The following specific comments should be addressed by the authors.

Major Comments

1. For clarity, the authors should always refer to "termination" and "termination efficiencies" as "translation termination" and "translation termination efficiencies". Currently, when the term "termination" is used in gene expression, it almost always refers to transcriptional termination. Particularly for the sake of readers who are reading the abstract for the first time, it's important to clearly state the study's results by adding this clarification. Within the manuscript, the authors also refer to "terminators" as the genetic part that terminates translation. Instead, the authors should refer to this genetic part as the stop codon.
2. The authors make two BIG assumptions that have an outsized impact on all of their results. First, they assume that all RNAs have a fixed degradation rate of 0.0067 1/sec (1.7 minute half-life). Second, they assume that all protein coding sequences have the same translation elongation rate. Neither of these assumptions are correct or valid, and the authors hand-waive them away by citing some of the original RiboSeq papers (e.g. Li et. al. 2014) that also do not provide any support of these assumptions. These assumptions have a HUGE effect on the authors' calculations and results. It's therefore disappointing that the authors claim to calculate absolute translation rates, but do not directly address what they (probably) know is a central deficiency of the RiboSeq technique that could prevent them from obtaining an accurate answer. The authors can remedy the situation with the following analyses:
3. Instead of assuming that all RNAs have the same degradation rate, the authors can readily use pre-existing measurements of mRNA decay rates in *E. coli*, for example, from [Chen, Huiyi, et al. "Genome-wide study of mRNA degradation and transcript elongation in *Escherichia coli*." *Molecular systems biology* 11.1 (2015): 781.], [Bernstein, Jonathan A., et al. "Global analysis of mRNA decay and abundance in *Escherichia coli* at single-gene resolution using two-color fluorescent DNA microarrays." *Proceedings of the National Academy of Sciences* 99.15 (2002): 9697-9702.] The importance of non-constant mRNA decay rates can not be over-stated. As described in the latter report, "A wide range of stabilities was observed for individual mRNAs of *E. coli*, although $\approx 80\%$ of all mRNAs had half-lives between 3 and 8 min." That means that, even for those 80% of mRNAs, the authors' calculated transcription rates are inaccurate by a large amount. For example, if we assume an even mRNA decay distribution between 3 and 8 minutes, the transcription rates will be inaccurate by 5.5-fold (on average). This improved analysis will also have an effect on the authors' comparison between transcription and translation rates across the *E. coli* genome.
4. Instead of assuming that all protein coding sequences have the same translation elongation rate, the authors could use prior measurements of ribosome-codon dwell times, for example, available at [Fluitt, Aaron, Elsje Pienaar, and Hendrik Viljoen. "Ribosome kinetics and aa-tRNA competition determine rate and fidelity of peptide synthesis." *Computational biology and chemistry* 31.5-6 (2007): 335-346.]. Based on this report, a ribosome's translation elongation rate varies by over 10-fold across different codons, making this an especially large source of inaccuracy in the authors' calculated translation rates. The authors can not claim that their calculations yield absolute translation rates if they don't account for differences in translation elongation rate across different genes.
5. More specifically, the authors state that "If we assume that each ribosome translates at a relatively

constant speed, which holds true in most cases (Gorochowski et al, 2015; Li et al, 2014), then the RPF coverage is proportional to the number of ribosomes at each nucleotide at a point in time and thus captures relative differences in ribosome flux; more heavily translated regions will have a larger number of ribosomes present and so accrue a larger number of RPF reads in the Ribo-seq snapshot." This is NOT generally true as shown by the report above. It is also contradicted by the general need to carry out synonymous codon optimization to introduce 'fast' codons into a protein coding sequence to increase its translation elongation rate. The authors can not make this assumption so lightly.

6. More clarification is also needed here: The authors state that "To determine whether translation rates were constant across each gene, we compared the number of RPFs mapping to the first and second half of each coding region." However, these criteria does NOT test whether all protein coding sequences have the same translation elongation rate. Instead, these criteria ONLY tests whether the first and second halves of a coding sequence have the same translation elongation rate. From an evolutionary perspective, any coding sequence that failed these criteria would be suboptimal, for example, by sequestering many ribosomes or leaving large portions of the mRNA unprotected by ribosomes. Therefore, it should not be surprising that *within the same CDS* the translation elongation rate is mostly constant. However, *across different CDSs* the translation elongation rates are not constant. The authors need to modify their text to clarify this difference.

7. More methodological analysis is also needed to support the authors' calculations. Some key questions include: How many ribosome protected fragments (RPFs) are measured in regions where no translation is expected to occur? This could be considered the background RPF level. What is the lowest RBF detection limit as compared to the spike-in RNA controls? How do these two numbers compare? When we see RBFs upstream of an RBS, how could one distinguish this as artificial background versus an actual ribosome translation an upstream region? The background and detection levels will influence the corrections terms calculated by the authors, e.g. $C(x)$. What are the typical values of $C(x)$ compared to the calculated ribosome fluxes $R(x)$? Are these small or big correction terms?

8. Regarding the RNA-Seq measurements, how much did mRNA levels change across all genes? In Li et. al. 2014, mRNA levels varied a great deal more than RBF density levels. From those measurements, it appeared that expression control was due to changes in transcription rate and less so by translation rate. How do your measurements compare? What is the overall range and distribution of mRNA levels compared to RBF levels across all genes?

9. Regarding the pseudoknot characterization, is there a stop codon in-frame with gene10? Is it possible that the RBFs downstream of gene10 are caused by incomplete translation termination from the stop codon? When analyzing RBF read sequences, how would one differentiate between incomplete translation termination and translation frameshifting? Is it simply a 1-nucleotide shift in the P-site position within the read? Is there any variation in the P-site nucleotide's position within the read? What is the distribution of these positions across all RBF reads?

10. Related to #9, the authors should show the P-site position distribution from their reads to illustrate how they determine ribosome codon-specific occupancies. The methodology is described in the authors' methods section, but a supporting figure (could be supplementary) would be useful.

11. Regarding the pseudoknot characterization, it is unclear how to visually interpret ribosome flux vs. nucleotide position plots as the shift in the 3-nucleotide periodicity is hidden by the calculation for ribosome flux. It would be better to calculate and show the ribosome flux vs. position in all three frames (using -1, 0, +1 shifts to the RBF's P-site). Then the reader could see the change in ribosome flux in all three open reading frames versus nucleotide position.

Minor Comments:

12. In their results section, the authors mention that, in eukaryotic translation, "In this case, no ribosome flux is generated by upstream genes." It is not clear why this paragraph is present in the results as all of their results are specific to prokaryotic translation. However, in eukaryotes, there are translated upstream open reading frames (uORFs). A fraction of the ribosomes that translation uORFs will dissociate and reassemble at downstream ORFs. This is a mechanism responsible for

translation regulation in eukaryotes. Again, this topic is not relevant to the authors' results and should not be included in their results section.

13. On line 392, the authors mention that "we next computed the dwell time of ribosomes at each codon". However, there is no time measurement here, particularly as the authors state that all measurements are made under steady-state conditions. The authors should refer to this as ribosome occupancy for clarity.

14. There are a few places where the authors describe their data analysis as "biophysical models". This seems like an incongruous term. What are the physics involved in this data analysis? All of the equations assume steady-state, resulting in simple proportional statements that do not invoke any physical phenomenon. There is no attempt to develop a model to explain why some mRNAs have higher or lower translation initiation rates or translation termination efficiencies. The authors should dial back the description of their work as biophysical modeling.

Reviewer #3:

The authors report experimental and computational methods for absolute quantification of protein synthesis in *E. coli*. They apply their methodology to analyze the transcriptional- and translational-level responses to stress induced by pseudoknot sequestration of ribosomes. The topic of this study is important, but the novelty of the methodology described here is questionable as are many of the key assumptions of the underlying model.

Major comments:

1) The authors' model assumes that transcript half-lives are constant (invariant across genes). However, the genome-wide distributions of RNA degradation have been characterized in *E. coli* by RNA-seq in previous studies (e.g. Chen et al, *Molecular Systems Biology*, 2015) in multiple growth phases. These studies show that RNA degradation rates vary over an order-of-magnitude. The authors should characterize the impact of this assumption on their model in light of these previous measurements. To what extent does the genome-wide variation in RNA stability impact the authors' quantification method? If this assumption is poor, couldn't the authors employ the straightforward RNA-seq methodology in Chen et al to avoid making this assumption and directly measure genome-wide half-lives or use the data provided by Chen et al?

2) The authors state that "more heavily translated regions will have a larger number of ribosomes present and so accrue a larger number of RPF reads in the Ribo-seq snapshot". While I agree that a high average ribosome density on a transcript is generally indicative of a high level of translation of that transcript, this interpretation depends entirely on the positional resolution with which one is examining ribosome density. At the single-codon level, fluctuations in ribosome density are generally thought to indicate ribosomal stalling (and therefore reduced translation rate). It seems that the authors are analyzing RPF coverage $N(x)$ where x is a single nucleotide position, but the interpretation that higher ribosome density at a given position x indicates higher levels of translation seems very problematic. Indeed, many prior studies interpret higher RPF coverage at a position x relative to other positions on the same transcript to indicate ribosomal stalling (e.g. Zhang et al, *Cell Systems*, 2017; Woolstenhulme et al, *Cell Reports*, 2015; Li et al, *Cell*, 2014), which results in an aberrant build-up of ribosome density at the codon where the ribosome is stalled.

3) The authors claim to assess their assumption that "ribosomes traverse the coding sequence at constant speed" by determining whether "the two halves of a transcript... have a near identical RPF coverage". First, while I agree that this a necessary condition for validating the assumption, it is not sufficient because there can be large fluctuations in ribosome density at the single-codon level. Second, what the authors actually state is that they "found a high correlation between both halves". This observation seems unrelated to the test proposed by the authors. For example, if the first half of all transcripts had exactly 10% of the RPF coverage of the second half of all transcripts, then the two halves would be perfectly correlated, but the two halves would differ in RPF coverage by a factor of ten. I cannot tell from looking at Supplementary Figure S3 whether or not the two halves "have a near identical RPF coverage", mainly because the plot is on a log-scale and so small variations

correspond to factors of two or more. Perhaps the authors could actually show the distribution of ratios between the two halves and use a statistical test to show that there is not a significant deviation from a ratio of one.

4) In the discussion, the authors claim to "present a new approach to quantify transcription and translation in living cells" using a "modified version of RNA-seq and Ribo-seq". As far as I can tell, the authors made no modification to Ribo-seq and simply followed the standard protocol of Guo et al, Nature, 2010. For RNA-seq, they simply included spike-in RNAs in their library preparation to facilitate normalization. This is common and has been discussed in detail in the literature in numerous papers (e.g. Lun et al, Genome Research, 2017; Jiang et al; Genome Research, 2011). Absolute quantification of protein synthesis by deep sequencing has been claimed in previous studies of E coli (e.g. Li et al; Quantifying absolute protein synthesis rates reveals principles underlying allocation of cellular resources; Cell; 2014). The authors claim in their discussion that the main difference between the current study and Li et al is the use of synthetic spike-in standards which enable the calculation of a detection limit. It is unclear to me the extent to which this advance really enables novel insight into translational regulation that would not be available using the methods reported by Li et al.

Minor comments:

1) The authors' model relies significantly on the steady-state assumption. To what extent is this assumption satisfied in the IPTG-induction experiments? My concern is that the onset of this induction will drive the cells away from a steady-state in the short-term. Do the authors have a way to assess this?

1st Revision - authors' response

7th March 2019

Reviewer #1:

This manuscript presents a quantitative analysis of mRNA abundance and translation in bacteria. Absolute mRNA abundances are estimated based on spike-in controls, whereas translation levels are calibrated based on steady-state protein levels. These calibrated data are used to describe the range of initiation and elongation rates for endogenous genes, as well as an inducible lacZ representative of heterologous protein expression. Having found that transcriptional induction of lacZ has little impact on its translation, the manuscript next analyzes a pseudoknot-dependent frameshift derived from bacteriophage. The frameshift is apparent in ribosome occupancy profiles of the transcript, and also induces broader physiological changes in cells that are attributed to sequestration of the translational machinery and the accumulation of unfolded or misfolded nascent proteins.

The absolute expression estimates presented in this manuscript are calibrated indirectly, depend on assumptions that are likely violated for many genes, and are not tested against any orthogonal measures. As such, the impact of these estimates over uncalibrated relative expression profiling are limited, and these fundamental caveats are not addressed. The cellular impact of pseudoknot overexpression is interesting, and may guide us to a deeper understanding of the fitness costs in heterologous protein expression. The current manuscript proposes interesting hypotheses about this effect, but does not test any of them.

We are grateful to the Reviewer for their detailed assessment of our work and are glad that they recognized the interesting and important questions we are attempting to address. In the revised manuscript, we have addressed Reviewer's concerns regarding the calibration of the measurements and clarify the transcriptional and translational response we observe during stress.

My major concerns with the manuscript:

1. The calibration of absolute mRNA abundance based on spike-in controls is plausible, but these are then converted to absolute initiation rates with a very simplistic estimate that all mRNA decay rates are constant and equal. This is unrealistic, and limits the value of these estimates.

We agree with the Reviewer that the assumption of a constant mRNA degradation rate for all genes will impact the accuracy of our measurements, especially as they are known to vary over an order of magnitude. Reflecting on this comment and taking the advice of Reviewer 2, we have incorporated mRNA specific degradation rates into our model using values from Chen *et al*, *Mol Syst Biol*, 2015. All performance calculations of genetic parts and associated figures have been updated throughout the manuscript.

2. *The calibration of absolute translation is even more limited. Absolute abundances are calibrated against total cellular protein content, dependent on the assumption that protein degradation is not a major factor in overall protein abundance. Again, this limits the value of the absolute estimates presented here.*

We respectfully disagree that our calculations of translation rates are limited by our calibration to total cellular protein content, and specifically that differences in protein degradation cast doubt on our measurements. It has been shown that >93% of the *E. coli* proteome is not subject to rapid degradation (both during exponential growth and even starvation conditions, see Nath & Koch, *J Biol Chem*, 1970). Most protein half-lives are well beyond the cell doubling time and so dilution by growth (an assumption our method relies on) is the major determinant of protein degradation rate. The Reviewer's concern may have arisen from a lack of clarity when presenting these previous experimental results, and so the "Generating transcription and translation profiles in absolute units" section has been updated to expand upon these points and include a supporting citation.

3. *No data is presented calibrating transcription or translation initiation rate estimates against any sort of true rate measurement (versus inference based on steady-state levels).*

There are some examples of precise *in vitro* measurements of transcription and translation initiation based on single gene/transcript studies (Andreeva *et al*, *PNAS*, 2018; Iyer *et al*, *Nucleic Acids Res*, 2016; Kennell and Riezman, *J Mol Biol*, 1977; Muthukrishnan *et al*, *PLoS One*, 2014; Petrov *et al*, *CSH Perspect Biol*, 2012; Volkov *et al*, *Nat Chem Biol*, 2018), but rates derived in this setting may differ several orders of magnitude compared to *in vivo* conditions. Instead, we chose to compare our measurements to those from a previous study (see Figure 3A) that showed a close correspondence to quantitative proteomics using mass spectrometry for a large subset of genes. These measurements do rely on the cells having reached steady state during exponential growth, but this assumption is fair, especially given the fact that the majority of the proteome is long-lived under a broad range of conditions (Nath & Koch, *J Biol Chem*, 1970; also see Comment 2 above). Large deviations in degradation rates will be rare and thus only have a small effect on a few of the synthesis rates inferred. Furthermore, the rates we obtain for the initiation rates are in very good agreement with another study of mRNA and protein synthesis rates (Kennell & Riezman, *J Mol Biol*, 1977). Specifically, Kennell & Riezman (*J Mol Biol*, 1977) find for the *lac* operon initiation rates of the P_{lac} promoter to be ~0.33 RNAP/s and a translation initiation rate of 0.08 ribosomes/s. These values closely match our measured transcription initiation rate of 0.3 RNAP/s for P_{lac} and an average translation initiation rate of across the genome of 0.18 ribosomes/s.

Given the Reviewers comment, we have realized that the initial submission did not thoroughly describe the existing literature in this area or explain how the rates we calculate closely match those of related genetic parts measured in alternative contexts. To address this, we have added further information to the "Measuring genome-wide translation initiation and translation termination in Escherichia coli" and "Quantifying differences in transcription and translation of endogenous and synthetic genes" sections and included additional citations.

4. *The manuscript presents an interesting analysis of relative transcription and translation rates (although limited as discussed above). However, these are interpreted as indicating genes that are "mostly governed by translation" versus those that are "mostly controlled by transcription". In the single, steady-state condition here, steady-state abundance is just the product of these two initiation rates and it isn't clear how to meaningfully partition "control". Control implies the extent to which changes in protein abundance result from changes in these synthesis rates (or decay rates, which*

are also regulated). At most this analysis can describe the range of initiation rates observed across different genes.

This Reviewer makes an excellent point and we regret having not been clearer with our wording. It was not our intention to suggest that overall protein levels are fully controlled by synthesis rates alone, as degradation obviously plays a crucial role. To address this concern, we have revised these sentences to explicitly state that we are comparing the relative contributions of transcription (mRNA synthesis rates and copy numbers) and translation (protein synthesis rates) across all genes. These changes do not affect our findings in this section, i.e. highly expressed endogenous proteins tend to have high mRNA synthesis rates/copy numbers and relatively low translation initiation rates.

5. *The manuscript reports that, "the PK caused 2-3% of ribosomes to frameshift, ~3-fold less than the 10% reported for the PK in its natural context". This should be verified by protein-level measurements - perhaps the quantitative estimate here (or in Condrón et al) is simply incorrect?*

This Reviewer is noticing the relatively low frameshifting efficiency we calculated based on the RPF reads as compared to what Condrón et al. presented in their original paper on characterizing the PK10 frameshifting efficiency. We used a strain overexpressing PK10 which originates from later work from the Michael Sorensen's group (Tholstrup et al., *Nucl Acid Res* 42, 2012). In this later publication, Tholstrup and coauthors produced various variants of PK10 which differ in their ability to stall the ribosomes and to frameshift. As we have mentioned in the Material and Methods section, we use variant 22/6a which is a variant of the natural PK10 the authors produced. Variant 22/6a has much lower frameshifting compared to the natural gene10 in the bacteriophage but exhibits a much higher efficiency in sequestering ribosomes. The latter was of importance for our study, hence the choice. Unlike the wild-type PK10 whose frame-shift efficiency reaches 10% (Fig. 1, Condrón et al., *J Bact* 1991), the efficiency of the 22/6a variant is 3% measured on a protein level, by incorporating radioactive Met (Fig. 3B, Tholstrup et al., *Nucleic Acid Res* 42, 2012). This closely matches our sequencing-based measurement of 2-3%.

We apologize for not having included a thorough explanation of our choice of the PK10 variant and not having emphasized on the differences with the natural PK10 gene described in Condrón et al. (*J Bact* 173, 1991). In the revised manuscript we include an explanation for our choice of the construct, emphasize its enhanced ability to stall ribosomes, and decreased frameshifting efficiency. We also include information about the quantitative differences between our construct and the wild-type PK10 and highlight the similarity of our measurement of frameshifting efficiency to those using Met-incorporation (Tholstrup et al., *Nucl Acid Res* 42, 2012) and RPFs from our sequencing data sets.

6. *The manuscript reports, "...a large number of RPF reads within the gene10 region...many of these reads capture stalled ribosomes." What is the evidence or argument supporting this interpretation?*

We measure a large drop of 80–90% in the translation profile directly after the pseudoknot. This could be caused by either premature termination of the ribosome or stalling to enrich the counts in the *gene10* region. The pseudoknot we use (variant 22/6a) has been experimentally shown to efficiently stall ribosomes (Tholstrup et al., *Nucleic Acid Res* 42, 2012) making it very likely that many of the RPF reads capture this feature. We realize that only limited information about the function and previous experimental characterization of the specific pseudoknot we chose was given in our initial submission. Therefore, building on Comment 5 above, previous experimental evidence have been provided in the "Cellular response to a strong synthetic pseudoknot" section, including a citation to Tholstrup et al. (*Nucleic Acid Res* 42, 2012).

Reviewer #2:

The authors utilize RNA-Seq and Ribo-Seq together to measure mRNA levels and ribosome densities in engineered E. coli strains together with spike-in RNA controls. In one engineered strain, a synthetic promoter is used to inducibly express lacZ. In another engineered strain, a natural RNA

pseudoknot within a viral gene10 is introduced upstream and out-of-frame of the lacZ coding sequence, creating the potential for translation frameshifting. By applying advanced data analysis, the authors use their measurements to calculate the apparent translation initiation rates at start codons and the apparent translation termination rates at stop codons. They use these calculations together with spike-in RNA controls, total protein mass measurements, and per-protein mass calculations to estimate the absolute ribosome fluxes (ribosomes per second) across transcripts. By comparing RiboSeq and RNA-Seq measurements across their engineered strains, the authors identify significant changes in transcription and translation rates across many genes, likely due to the pseudoknot-mediated frameshifting and the resulting excess sequestration of ribosomes. Overall, the work presented is high-quality and interesting, demonstrating how RiboSeq and RNA-Seq can be used to quantify transcriptome and proteome-wide changes in cell physiology. However, the authors do their best to explain away some assumptions that could significantly alter their calculated results. The RiboSeq technique is fairly new and not without bias. The following specific comments should be addressed by the authors.

We are pleased that this Reviewer recognized the quality of our work and found it interesting. We are also very grateful for their helpful suggestions regarding new analyses and modifications to the models. These have been incorporated into the updated manuscript, which we believe is now greatly improved.

Major Comments

1. For clarity, the authors should always refer to "termination" and "termination efficiencies" as "translation termination" and "translation termination efficiencies". Currently, when the term "termination" is used in gene expression, it almost always refers to transcriptional termination. Particularly for the sake of readers who are reading the abstract for the first time, it's important to clearly state the study's results by adding this clarification. Within the manuscript, the authors also refer to "terminators" as the genetic part that terminates translation. Instead, the authors should refer to this genetic part as the stop codon.

We apologise for any confusion this might have caused. As recommended by the Reviewer, we have updated the manuscript to be explicit when mentioning “termination” in different contexts (i.e. transcription and translation) and used “stop codon” to refer to the genetic part terminating translation.

2. The authors make two BIG assumptions that have an outsized impact on all of their results. First, they assume that all RNAs have a fixed degradation rate of 0.0067 1/sec (1.7-minute half-life). Second, they assume that all protein coding sequences have the same translation elongation rate. Neither of these assumptions are correct or valid, and the authors hand-waive them away by citing some of the original RiboSeq papers (e.g. Li et. al. 2014) that also do not provide any support of these assumptions. These assumptions have a HUGE effect on the authors' calculations and results. It's therefore disappointing that the authors claim to calculate absolute translation rates, but do not directly address what they (probably) know is a central deficiency of the RiboSeq technique that could prevent them from obtaining an accurate answer. The authors can remedy the situation with the following analyses:

We agree that the models presented in the original submission made assumptions regarding both mRNA degradation and translation elongation rates that would reduce the accuracy of our measurements. This is a fair criticism of our approach and thus as advised by the Reviewer, we have updated our models to incorporate more detailed data regarding each of these aspects (see also our responses to Comments 3 and 4 below).

3. Instead of assuming that all RNAs have the same degradation rate, the authors can readily use pre-existing measurements of mRNA decay rates in E. coli, for example, from [Chen, Huiyi, et al. "Genome-wide study of mRNA degradation and transcript elongation in Escherichia coli." Molecular systems biology 11.1 (2015): 781.], [Bernstein, Jonathan A., et al. "Global analysis of mRNA decay and abundance in Escherichia coli at single-gene resolution using two-color fluorescent DNA microarrays." Proceedings of the National Academy of Sciences 99.15 (2002): 9697-9702.] The importance of non-constant mRNA decay rates cannot be over-stated. As described in the latter report, "A wide range of stabilities was observed for individual mRNAs of E.

coli, although $\approx 80\%$ of all mRNAs had half-lives between 3 and 8 min." That means that, even for those 80% of mRNAs, the authors' calculated transcription rates are inaccurate by a large amount. For example, if we assume an even mRNA decay distribution between 3 and 8 minutes, the transcription rates will be inaccurate by 5.5-fold (on average). This improved analysis will also have an effect on the authors' comparison between transcription and translation rates across the *E. coli* genome.

We gratefully acknowledge this suggestion. In the revised version we integrated mRNA specific degradation rates from Chen *et al*, *Mol Syst Biol*, 2015 into our model to improve the estimates of our transcription rates. These updates have been described in the section "Generating transcription and translation profiles in absolute units" and all part performance measurements have been recalculated (e.g. Figure 3).

4. Instead of assuming that all protein coding sequences have the same translation elongation rate, the authors could use prior measurements of ribosome-codon dwell times, for example, available at [Fluitt, Aaron, Elsje Pienaar, and Hendrik Viljoen. "Ribosome kinetics and aa-tRNA competition determine rate and fidelity of peptide synthesis." *Computational biology and chemistry* 31.5-6 (2007): 335-346]. Based on this report, a ribosome's translation elongation rate varies by over 10-fold across different codons, making this an especially large source of inaccuracy in the authors' calculated translation rates. The authors cannot claim that their calculations yield absolute translation rates if they don't account for differences in translation elongation rate across different genes.

We again acknowledge this Reviewer's suggestion and have incorporated this recommendation into our model. Now, in addition to distributing the total protein mass per cell to each gene according to their RPF densities, we also weight these values to account for differences in expected translation rate based on estimated codon translation times. This allows us to more accurately capture changes in elongation rate between coding regions. This change is described in the revised section "Generating transcription and translation profiles in absolute units" and include a new citation to Fluitt *et al*, *Comp Biol Chem*, 2007. In addition, all part performance measurements have been updated using this methodology (Figures 2–5 and Datasets EV1 and EV2).

5. More specifically, the authors state that "If we assume that each ribosome translates at a relatively constant speed, which holds true in most cases (Gorochowski *et al*, 2015; Li *et al*, 2014), then the RPF coverage is proportional to the number of ribosomes at each nucleotide at a point in time and thus captures relative differences in ribosome flux; more heavily translated regions will have a larger number of ribosomes present and so accrue a larger number of RPF reads in the Ribo-seq snapshot." This is NOT generally true as shown by the report above. It is also contradicted by the general need to carry out synonymous codon optimization to introduce 'fast' codons into a protein coding sequence to increase its translation elongation rate. The authors cannot make this assumption so lightly.

Given the changes we have made to the calculation of the translation profiles (incorporating codon specific differences in translation time, see Comment 4 above) our translation profiles will now be proportional to ribosome flux. We have reworded this entire section to clearly explain the changes made to the model to improve its accuracy.

6. More clarification is also needed here: The authors state that "To determine whether translation rates were constant across each gene, we compared the number of RPFs mapping to the first and second half of each coding region." However, these criteria does NOT test whether all protein coding sequences have the same translation elongation rate. Instead, these criteria ONLY tests whether the first and second halves of a coding sequence have the same translation elongation rate. From an evolutionary perspective, any coding sequence that failed these criteria would be suboptimal, for example, by sequestering many ribosomes or leaving large portions of the mRNA unprotected by ribosomes. Therefore, it should not be surprising that *within the same CDS* the translation elongation rate is mostly constant. However, *across different CDSs* the translation elongation rates are not constant. The authors need to modify their text to clarify this difference.

We apologise for having not clearly stated that our analysis only allows for verification of similar translation rates within a CDS and not across them. We have edited the section “Measuring genome-wide translation initiation and translation termination in *Escherichia coli*” to rectify this.

7. More methodological analysis is also needed to support the authors' calculations. Some key questions include: How many ribosome protected fragments (RPFs) are measured in regions where no translation is expected to occur? This could be considered the background RPF level. What is the lowest RBF detection limit as compared to the spike-in RNA controls? How do these two numbers compare? When we see RBFs upstream of an RBS, how could one distinguish this as artificial background versus an actual ribosome translation an upstream region? The background and detection levels will influence the corrections terms calculated by the authors, e.g. $C(x)$. What are the typical values of $C(x)$ compared to the calculated ribosome fluxes $R(x)$? Are these small or big correction terms?

The background RPF levels (i.e. those outside known and identified ORFs) were measured to be <0.003 RPFs per nucleotide per million mapped RPFs for all samples. This is well below the detection limit set on the RNA spike-ins and would have virtually no impact on the calculation of the RBS or stop codon performance. Similarly, The $C(x)$ correction term for the RBSs is typically very small when compared to $R(x)$. Specifically, 0.06% and 0.1% across all RBSs in cells harbouring the *lacZ* construct in the absence and presence of IPTG, respectively. To explain these points the section “Measuring genome-wide translation initiation and translation termination in *Escherichia coli*” was revised to include details regarding background RPF levels and the scale of the corrections for RBS measurements.

Even though the levels in the system we study are very low, it is important to recognise that engineered DNA sequences could include faults in their design leading to high levels of unwanted translation (e.g. due to cryptic genetic parts). Therefore, we consider inclusion of this correction term important for some systems.

8. Regarding the RNA-Seq measurements, how much did mRNA levels change across all genes? In Li et. al. 2014, mRNA levels varied a great deal more than RPF density levels. From those measurements, it appeared that expression control was due to changes in transcription rate and less so by translation rate. How do your measurements compare? What is the overall range and distribution of mRNA levels compared to RPF levels across all genes?

We have generated new plots showing the distribution of mRNA levels and normalised RPFs per gene (Figure S5 in revised Supplementary Appendix) and updated the section “Quantifying differences in transcription and translation of endogenous and synthetic genes” to introduce them. We find that Figure S5 clearly shows greater variation in RPF densities across the genome than mRNA copy numbers. It also illustrates that while most mRNAs are stochastically expressed at around ~0.1 copies per cell, there are also a small fraction with a much higher copy number (~60 copies per cell) corresponding to key cellular machinery (e.g. ribosomal RNA). Although this provides an overview of the variability under normal growth conditions, we do find that large changes in overall protein synthesis rates are mostly due to transcriptional regulation and specifically changes in mRNA copy numbers, as shown in Figure 5B and explained in the section “Cellular response to a strong synthetic pseudoknot”.

*9. Regarding the pseudoknot characterization, is there a stop codon in-frame with *gene10*? Is it possible that the RBFs downstream of *gene10* are caused by incomplete translation termination from the stop codon? When analyzing RBF read sequences, how would one differentiate between incomplete translation termination and translation frameshifting? Is it simply a 1-nucleotide shift in the P-site position within the read? Is there any variation in the P-site nucleotide's position within the read? What is the distribution of these positions across all RBF reads?*

There is an in-frame stop codon at the end of *gene10*, and as the Reviewer highlights, it is possible that some of the RPFs post *gene10* are due to incomplete termination. To ensure the influence of this read-through was not the major factor in our frameshift calculation, in the original submission we performed additional analysis of the reading frames for each RPF read in the *lacZ* gene region (downstream of *gene10*'s stop codon). This showed a large shift to out-

of-frame translation (−1 and +1 in Figure 4C) mostly into the −1 frame. As the fraction of in-frame RPFs is below the background levels we see for out-of-frame RPFs across all coding regions in the genome (Figure 4D), it is thus likely a very small contribution to overall ribosome flux. This is further supported by our accurate measurement of the PK frameshift efficiency, matching measurements of an identical construct using protein expression levels monitored by radioactive Met-incorporation (Tholstrup *et al.*, *Nucl Acid Res* 42, 2012).

To clarify these points, we have updated the section “Characterizing a synthetic pseudoknot that induces translational recoding” to emphasize the fact that the *gene10* coding region ends with a stop codon and expanded the discussion of Figure 4C so that the reader is aware of the shift in the reading frames inferred from the RPFs.

10. Related to #9, the authors should show the P-site position distribution from their reads to illustrate how they determine ribosome codon-specific occupancies. The methodology is described in the authors' methods section, but a supporting figure (could be supplementary) would be useful.

To help better illustrate our methodology, as recommended by this Reviewer, we have added Figure S6 to the Supplementary Appendix to show the method of determining the P site from each RPF read. We have also added Figure S5 to the Supplementary Appendix to show the codon occupancies for each construct and condition.

11. Regarding the pseudoknot characterization, it is unclear how to visually interpret ribosome flux vs. nucleotide position plots as the shift in the 3-nucleotide periodicity is hidden by the calculation for ribosome flux. It would be better to calculate and show the ribosome flux vs. position in all three frames (using -1, 0, +1 shifts to the RBF's P-site). Then the reader could see the change in ribosome flux in all three open reading frames versus nucleotide position.

Due to the relatively low translation rates after the pseudoknot, and especially throughout the *lacZ* coding sequence, noise in the frame-specific RPF profiles makes them difficult to compare. Instead, we opted to include a comparison of the fractions of RPFs for each frame across the various regions (see Figure 4C). This clearly shows the shifts in the major translation frame from being in-frame throughout *gene10* to frameshifted (mostly into the −1 frame) in *lacZ*. We fear that the original submission may not have explained these plots clearly enough and so have updated the “Characterizing a synthetic pseudoknot that induces translational recoding” section to further describe the frame-specific details of the RPF reads across the construct.

Minor Comments:

12. In their results section, the authors mention that, in eukaryotic translation, "In this case, no ribosome flux is generated by upstream genes." It is not clear why this paragraph is present in the results as all of their results are specific to prokaryotic translation. However, in eukaryotes, there are translated upstream open reading frames (uORFs). A fraction of the ribosomes that translate uORFs will dissociate and reassemble at downstream ORFs. This is a mechanism responsible for translation regulation in eukaryotes. Again, this topic is not relevant to the authors' results and should not be included in their results section.

We agree with the Reviewer that the paragraph related to eukaryotic translation may confuse a reader because of the inherent differences in the initiation mechanism between prokaryotes and eukaryotes. Therefore, as recommended by the Reviewer, we removed it from the manuscript.

13. On line 392, the authors mention that "we next computed the dwell time of ribosomes at each codon". However, there is no time measurement here, particularly as the authors state that all measurements are made under steady-state conditions. The authors should refer to this as ribosome occupancy for clarity.

Indeed, “ribosome occupancy” represents precisely what is measured by the approach and so as advised we have replaced “dwell time” with “ribosome occupancy”.

14. There are a few places where the authors describe their data analysis as "biophysical models". This seems like an incongruous term. What are the physics involved in this data analysis? All of the equations assume steady-state, resulting in simple proportional statements that do not invoke any physical phenomenon. There is no attempt to develop a model to explain why some mRNAs have higher or lower translation initiation rates or translation termination efficiencies. The authors should dial back the description of their work as biophysical modeling.

We now refer to our models as "mathematical models" throughout the manuscript to avoid any confusion.

Reviewer #3:

The authors report experimental and computational methods for absolute quantification of protein synthesis in E. coli. They apply their methodology to analyze the transcriptional- and translational-level responses to stress induced by pseudoknot sequestration of ribosomes. The topic of this study is important, but the novelty of the methodology described here is questionable as are many of the key assumptions of the underlying model.

We are very pleased to read that the Reviewer recognizes the importance of this work and thank them for their thoughtful comments and suggestions. In the revised manuscript, we have attempted to clarify and emphasize the novelty of our approach and address concerns regarding the data analysis.

Major comments:

1) *The authors' model assumes that transcript half-lives are constant (invariant across genes). However, the genome-wide distributions of RNA degradation have been characterized in E. coli by RNA-seq in previous studies (e.g. Chen et al, Molecular Systems Biology, 2015) in multiple growth phases. These studies show that RNA degradation rates vary over an order-of-magnitude. The authors should characterize the impact of this assumption on their model in light of these previous measurements. To what extent does the genome-wide variation in RNA stability impact the authors' quantification method? If this assumption is poor, couldn't the authors employ the straightforward RNA-seq methodology in Chen et al to avoid making this assumption and directly measure genome-wide half-lives or use the data provided by Chen et al?*

This point was also raised by the other two Reviewers (Reviewer 1, Comment 1 and Reviewer 2, Comment 2) and we have addressed it by updating our model to include mRNA specific degradation terms (taken from Chen et al., *Mol Syst Biol*, 2015). Details are provided in the revised "Generating transcription and translation profiles in absolute units" section.

2) *The authors state that "more heavily translated regions will have a larger number of ribosomes present and so accrue a larger number of RPF reads in the Ribo-seq snapshot". While I agree that a high average ribosome density on a transcript is generally indicative of a high level of translation of that transcript, this interpretation depends entirely on the positional resolution with which one is examining ribosome density. At the single-codon level, fluctuations in ribosome density are generally thought to indicate ribosomal stalling (and therefore reduced translation rate). It seems that the authors are analyzing RPF coverage $N(x)$ where x is a single nucleotide position, but the interpretation that higher ribosome density at a given position x indicates higher levels of translation seems very problematic. Indeed, many prior studies interpret higher RPF coverage at a position x relative to other positions on the same transcript to indicate ribosomal stalling (e.g. Zhang et al, *Cell Systems*, 2017; Woolstenhulme et al, *Cell Reports*, 2015; Li et al, *Cell*, 2014), which results in an aberrant build-up of ribosome density at the codon where the ribosome is stalled.*

We thank the Reviewer for raising this important point and believe that their concerns were raised from our somewhat unclear description about how we use the raw translation profiles. It is true that large fluctuations in ribosome densities can occur due to transient pausing/stalling of the ribosomes caused by features like rare codons, specific amino acids

interacting with the tunnel entrance, difficult to translocate codons with specific amino acid signature, evolutionarily selected secondary features as plentiful published examples including our previous publications describe. While important biologically, such events are rare (e.g. 89 stalling sites with strong secondary structure throughout all coding regions in *E. coli* were identified; Del Campo *et al.*, *PLoS Genetics*, 2015), and so their effect can be mitigated by considering an average of the translation profile in a region of interest. This is precisely what we do when calculating part performance (see Equations 2, 6 and 7). We also carefully check that RPF densities are similar for the 1st and 2nd halves of each gene (Figure S3), to verify that rare localised stalling effects do not majorly impact the average density calculations. To highlight the impact of these events and how they are handled in our approach, we have updated the “Generating transcription and translation profiles in absolute units” section to explain the effect of these processes and included additional citations (provided by the Reviewer).

3) *The authors claim to assess their assumption that "ribosomes traverse the coding sequence at constant speed" by determining whether "the two halves of a transcript... have a near identical RPF coverage". First, while I agree that this a necessary condition for validating the assumption, it is not sufficient because there can be large fluctuations in ribosome density at the single-codon level. Second, what the authors actually state is that they "found a high correlation between both halves". This observation seems unrelated to the test proposed by the authors. For example, if the first half of all transcripts had exactly 10% of the RPF coverage of the second half of all transcripts, then the two halves would be perfectly correlated, but the two halves would differ in RPF coverage by a factor of ten. I cannot tell from looking at Supplementary Figure S3 whether or not the two halves "have a near identical RPF coverage", mainly because the plot is on a log-scale and so small variations correspond to factors of two or more. Perhaps the authors could actually show the distribution of ratios between the two halves and use a statistical test to show that there is not a significant deviation from a ratio of one.*

We apologise for having not clearly presented this important aspect of the data in the initially submitted manuscript. To address this issue, Figure S3 has been updated to show new plots of the log₂ fold-change between RPF counts for the 1st and 2nd half of each coding region. In the case of a perfect agreement between both halves, a value of 0 would be given. These new plots show a small deviation of less than ±1.5-fold for 80% of genes, suggesting that ribosome speed is fairly constant across each coding region.

4) *In the discussion, the authors claim to "present a new approach to quantify transcription and translation in living cells" using a "modified version of RNA-seq and Ribo-seq". As far as I can tell, the authors made no modification to Ribo-seq and simply followed the standard protocol of Guo *et al.*, *Nature*, 2010. For RNA-seq, they simply included spike-in RNAs in their library preparation to facilitate normalization. This is common and has been discussed in detail in the literature in numerous papers (e.g. Lun *et al.*, *Genome Research*, 2017; Jiang *et al.*; *Genome Research*, 2011). Absolute quantification of protein synthesis by deep sequencing has been claimed in previous studies of *E. coli* (e.g. Li *et al.*; *Quantifying absolute protein synthesis rates reveals principles underlying allocation of cellular resources*; *Cell*; 2014). The authors claim in their discussion that the main difference between the current study and Li *et al.* is the use of synthetic spike-in standards which enable the calculation of a detection limit. It is unclear to me the extent to which this advance really enables novel insight into translational regulation that would not be available using the methods reported by Li *et al.**

The Reviewer is correct in stating that the experimental sequencing methods we choose have been used and validated in other studies. The modification we were referring to in this work was our integration of these previously separated approaches into a single coherent methodology, which to the best of our knowledge is unique for this study. This allows us to provide complementary information to characterise genetic parts regulating both transcription and translation simultaneously and to crucially measure key attributes (i.e. RNAP and ribosome flux) in absolute units. No previous studies have been able to provide such a complete and comprehensive picture of these central processes – a point we have clearly not articulated in our initially submitted version. Moreover, vital to these measurements is the interpretation of combined sequencing data using new mathematical models derived in this work. These are a clear novel contribution and lay an important foundation for better

understanding the function of gene regulatory components – not merely their combined effect on protein synthesis rate.

In regard to the concern about novel insight going beyond the previous work of Li *et al.* (Cell, 2014), we would like to stress that Li *et al.* could only assess overall protein synthesis rates of each gene under the assumption of uniform translation elongation rates across the transcriptome. Our updated method relaxes this assumption (see Reviewer 2, Comment 4) and by using absolute mRNA copy numbers rather than relative measurements of mRNA concentrations (i.e. RPKMs), we are able to capture synthesis rates per mRNA (e.g. see Figure 3A). This difference may seem slight; however, being able to assess biological processes in absolute units is vital, if we are to be able to validate our mechanistic understanding and integrate known physical limitations and constraints in these processes. The limited use of absolute units across biology has been raised as a significant concern (Justman, *Cell Systems*, 2018), which hinders our ability to reason about biological systems and apply rigorous mathematics. Another major limitation of using relative units as in Li *et al.* (Cell 2014) is that they severely limit data reuse. For example, we may be interested to know how the strength of particular regulatory elements differs across microbes. Using measurements in relative units would not allow this; comparison of promoter transcription strengths in say RPKM units is only possible if the total concentration of RNA per cell remains similar between samples. This will breakdown when comparing across species (e.g. bacteria vs. yeast). Because our work generates measurements in absolute units it overcomes all of these limitations, allowing broad reuse and direct comparisons.

To address this Reviewer's concerns and clarify the novel features of our approach, we have made changes to the Abstract, Introduction and Discussion, as well as including a new Synopsis to help better summarise our contributions in the context of previous work and the novel directions and avenues that it opens.

Minor comments:

1) *The authors' model relies significantly on the steady-state assumption. To what extent is this assumption satisfied in the IPTG-induction experiments? My concern is that the onset of this induction will drive the cells away from a steady-state in the short-term. Do the authors have a way to assess this?*

The 10–15 min induction time we use is sufficient for most mRNA levels to reach steady state due to their native turnover rate (i.e. average mRNA lifetime ~6 min, see Chen *et al.*, *Mol Syst Biol*, 2015). The available translational resources (i.e. ribosomes) will not see a significant change in their concentration during this short period, meaning the footprints we measure after induction will come from a redistribution of these existing translation resources. Because Ribo-seq is a snapshot of ribosome densities and not overall protein levels (i.e. we are measuring the process not the product), it does not require protein concentrations to have reached a steady state to enable a direct comparison of normalized RPF densities. The key requirement is that the concentration of translational resources does not change extensively across conditions (as this will affect the total ribosome flux that can be achieved across the transcriptome). In support of this, we measure virtually identical rates after induction of the LacZ construct that causes only a minor stress to the cell, and a uniform drop in rates across the entire transcriptome of RBS initiation rates for the PK construct, which sequesters a large proportion of the cellular ribosome pool diverting it uniformly from all endogenous genes.

2nd Editorial Decision

3rd April 2019

Thank you for sending us your revised manuscript. We have now heard back from the three reviewers who were asked to evaluate the revised work. As you will see below, the reviewers think that the study has significantly improved as a result of the performed revisions. However, reviewers #1 and #2 still raise some remaining issues, which we would ask you to address in a revision. All remaining issues can be addressed by text modifications and do not require further experimental or other analyses.

REFeree REPORTS

Reviewer #1:

The revisions have addressed some of my major concerns regarding the original submission. The incorporation of transcript-specific mRNA half-lives is a real improvement, and it's reassuring to see an agreement between frameshifting estimates here and protein-level measurements.

While the data and analysis are overall strong and allow robust, quantitative comparisons, I remain concerned about inferring specific rates in physical units (RNAP / second and so forth) with limited calibration or validation. I agree that these numbers are very hard to measure in cells -- but without this calibration data, there are a variety of ways these numbers might be inaccurate in absolute units despite being very biologically relevant and scientifically useful. It would be better to say that these physical rates are "inferred" or otherwise acknowledge the indirect nature of the estimates. It's also notable that some of the values reported here seem implausible -- initiation rates of 3.4 ribosomes / second, for instance, or a flux of 10 ribosomes / second across a codon.

Reviewer #2:

The authors have modified the analysis of their RNA-Seq and Ribo-Seq data to address this reviewers' comments. After re-reading the manuscript, the authors have greatly improved the descriptions of their analysis. However, the introduction is full of non-relevant material, seemingly disconnected from the authors' results, that seeks to push a particular ideology within the Synthetic Biology community. As this is a research article, and not a Perspective piece, readers would be better served if the authors focused their introduction on the most salient points connected to their results (more on this below).

Specific Comments:

1. The authors introduce the motivations to their work with the overall challenge of "predicting how a part will behave when assembled with many others" and "we have yet to reach a point where large and robust genetic circuits can be reliably built on our first attempt". These are worthy challenges, but the authors' measurements don't actually address them. First, there are no predictions here (either for genetic part function or genetic system function) and the authors don't make the connection to how their measurements will improve our ability to predict genetic part or genetic system function. Second, the genetic circuits tested here are very simple (an IPTG-inducible promoter controlling LacZ expression) and a similar genetic circuit utilizing a pseudoknot to create frameshifted expression of LacZ. Notably, the authors do not observe very large changes in transcription or translation rates in endogenous genes when LacZ expression is induced by IPTG. But they do observe large changes when a pseudoknot is introduced to cause frameshifting. These are interesting results, but how are they related to the central challenges stated in the authors' introduction? Most genetic circuits do not rely on pseudoknots for regulating gene expression.

2. The authors repeatedly refer to their work as measuring the "performance of parts" or the "high-throughput characterization of genetic parts". However, the vast majority of the authors' data are measurements of endogenous promoters and 5' untranslated regions (RBSs). There are distinct differences between what a Synthetic biologist would call a "genetic part" and the natural amalgam of gene regulatory signals that constitute an endogenous promoter or 5' UTR. First, most endogenous genes are transcribed by multiple overlapping promoters (ie, multiple transcriptional start sites). These promoters often have multiple overlapping signals and transcription factor binding sites, leading to complex transcriptional regulation. The authors can't distinguish the transcription rates from these multiple promoters, and they can not assign different RNAP/s absolute units to them. To compare, a Synthetic Biologist would call a single, well-defined promoter a "part" because they can move it around with (more-or-less) the same functionality. Measurements of endogenous promoters' transcription rates are not readily usable to develop an improved understanding (or predictions) of well-defined promoters as there are many confounding variables. The authors should rephrase their text (in several places) to distinguish between well-defined genetic parts and natural promoters.

3. Similarly, the authors should be aware that translation regulation is endemic across the transcriptome. Upstream and downstream coding sequences are co-regulated by translational coupling. Coupling between translation and mRNA stability exists. And some genes even utilize multiple in-frame start codons to express multiple protein isoforms. Again, the authors' measurements can't distinguish the regulated ways in which the ribosome bound to their mRNAs. Simply measuring "the translation rates" of these coding sequences doesn't necessarily improve our ability to understand these mechanisms, certainly not in a way that would enable us to predict genetic part or genetic system function as the authors describe in their introduction. This is another

reason why the authors need to tone down their perspective that NGS approaches will be the (one true) solution to the stated challenges (while dismissing other approaches).

4. The authors mention in their results that "For example, the most metabolically efficient way to strongly express a protein of interest in bacteria is by producing high numbers of transcripts (e.g. with high transcription initiation rate and high stability) with a relatively weak RBS (e.g. low translation initiation rate)." This is not true. Producing large numbers of mRNA transcripts is a metabolic cost that is not necessary if the mRNA can support both a high translation initiation and elongation rate without compromising protein folding. However, if the mRNA can't be recoded to support high translation elongation rates, or if the protein is prone to misfolding, due to ribosome-ribosome interactions, then the authors' strategy is a good one. But there are no generalities here. Only a minority of *E. coli* proteins undergo co-translational protein folding, and most of them involve membrane interactions (ie, not the transcription factors or enzymes commonly found in engineered genetic systems).

Reviewer #3:

In my opinion, the authors have done a very reasonable job of addressing my concerns and have significantly improved the discussion of their results and methodology. The incorporation of gene-specific degradation rates is major improvement.

2nd Revision - authors' response

9th April 2019

Reviewer #1:

The revisions have addressed some of my major concerns regarding the original submission. The incorporation of transcript-specific mRNA half-lives is a real improvement, and it's reassuring to see an agreement between frameshifting estimates here and protein-level measurements.

We thank again the Reviewer for their suggestions regarding the transcript specific degradation rates and other changes that we believe helped strengthen the work.

While the data and analysis are overall strong and allow robust, quantitative comparisons, I remain concerned about inferring specific rates in physical units (RNAP/second and so forth) with limited calibration or validation. I agree that these numbers are very hard to measure in cells -- but without this calibration data, there are a variety of ways these numbers might be inaccurate in absolute units despite being very biologically relevant and scientifically useful. It would be better to say that these physical rates are "inferred" or otherwise acknowledge the indirect nature of the estimates. It's also notable that some of the values reported here seem implausible -- initiation rates of 3.4 ribosomes / second, for instance, or a flux of 10 ribosomes / second across a codon.

We have made changes to the Introduction and the Discussion to clarify that the rates we calculate are "inferred". Specifically, in the Introduction we state: "We apply our method to *Escherichia coli* and demonstrate how local changes in these profiles can be interpreted using mathematical models to infer the performance of three different types of genetic part in absolute units.", and in the Discussion: "Because our methodology is based on sequencing, it can scale beyond the number of simultaneous measurements that are possible with common fluorescence-based approaches, and through the use of spike-in standards we are able to indirectly infer part parameters in absolute units (i.e. transcription and translation rates in RNAP/s and ribosomes/s units, respectively)."

In regard to the implausible initiation rates and ribosome fluxes, we cannot understand where the Reviewer is picking up these values. The genome analysis saw a maximum translation initiation rate of 1.8 ribosomes/s, which seems plausible given that the average rate in *E. coli* is thought to be ~0.2 ribosomes/s (Kennell & Riezman, 1977) and strong RBSs can easily boost gene expression several orders of magnitude beyond this. For the ribosome fluxes, the latest measurement of nascent chain elongation is 12 amino acids/s, which is in agreement with older estimates of 10-20 amino acids/s (Talkad V, Schneider E, Kennell D. *J Mol Biol* 104, 299–303, 1976) and would permit ribosomes fluxes of 10 ribosomes/s across a codon. However, none of our measurements reach this speed. Some confusion may have arisen from the fact that our translation profiles have a nucleotide resolution (3 times faster than per codon) and so to

clarify this we updated the “Generating transcription and translation profiles in absolute units” section to include: “*We next convert the weighted RPF coverage into a translation profile whose height corresponds directly to the ribosome flux across each nucleotide in ribosomes/s units.*”

Reviewer #2:

The authors have modified the analysis of their RNA-Seq and Ribo-Seq data to address this reviewers' comments. After re-reading the manuscript, the authors have greatly improved the descriptions of their analysis. However, the introduction is full of non-relevant material, seemingly disconnected from the authors' results, that seeks to push a particular ideology within the Synthetic Biology community. As this is a research article, and not a Perspective piece, readers would be better served if the authors focused their introduction on the most salient points connected to their results (more on this below).

We are delighted to read that the Reviewer appreciates the changes made to improve the methodology and the presentation of the results. As raised by the Reviewer, we have made major changes to the Introduction to focus it towards our results.

Specific Comments:

1. The authors introduce the motivations to their work with the overall challenge of "predicting how a part will behave when assembled with many others" and "we have yet to reach a point where large and robust genetic circuits can be reliably built on our first attempt". These are worthy challenges, but the authors' measurements don't actually address them. First, there are no predictions here (either for genetic part function or genetic system function) and the authors don't make the connection to how their measurements will improve our ability to predict genetic part or genetic system function. Second, the genetic circuits tested here are very simple (an IPTG-inducible promoter controlling LacZ expression) and a similar genetic circuit utilizing a pseudoknot to create frameshifted expression of LacZ. Notably, the authors do not observe very large changes in transcription or translation rates in endogenous genes when LacZ expression is induced by IPTG. But they do observe large changes when a pseudoknot is introduced to cause frameshifting. These are interesting results, but how are they related to the central challenges stated in the authors' introduction? Most genetic circuits do not rely on pseudoknots for regulating gene expression.

It was not the intention of this work to predict how genetic parts might behave when used in new ways. In fact, our motivation was to provide genome/circuit wide measurements of genetic parts such that a better understand of their function in a wide range of contexts would become possible. This in turn could then be used to refine and improve predictive models. To clarify this the Introduction has been updated to include: “A crucial step towards this goal will be to better understand how the many parts of large genetic circuits function in concert. However, approaches to simultaneously measure the performance of many parts within this context are currently lacking.”

2. The authors repeatedly refer to their work as measuring the "performance of parts" or the "high-throughput characterization of genetic parts". However, the vast majority of the authors' data are measurements of endogenous promoters and 5' untranslated regions (RBSs). There are distinct differences between what a Synthetic biologist would call a "genetic part" and the natural amalgam of gene regulatory signals that constitute an endogenous promoter or 5' UTR. First, most endogenous genes are transcribed by multiple overlapping promoters (ie, multiple transcriptional start sites). These promoters often have multiple overlapping signals and transcription factor binding sites, leading to complex transcriptional regulation. The authors can't distinguish the transcription rates from these multiple promoters, and they cannot assign different RNAP/s absolute units to them. To compare, a Synthetic Biologist would call a single, well-defined promoter a "part" because they can move it around with (more-or-less) the same functionality. Measurements of endogenous promoters' transcription rates are not readily usable to develop an improved understanding (or predictions) of well-defined promoters as there are many confounding variables. The authors should rephrase their text (in several places) to distinguish between well-defined genetic parts and natural promoters.

To recognise the differences between well-defined synthetic genetic parts and the endogenous sequences/elements controlling transcription and translation, the following changes were made to the Abstract: “*Here, we combine Ribo-seq with quantitative RNA-seq to measure at nucleotide resolution and in absolute units the performance of elements controlling transcriptional and translational processes during protein synthesis.*”, and Introduction: “*Here, we develop an approach that combines ribosome profiling (Ribo-seq) with quantitative RNA sequencing (RNA-seq) that enables the high-throughput characterization of endogenous sequences and synthetic genetic parts controlling transcription and translation in absolute units.*”

3. Similarly, the authors should be aware that translation regulation is endemic across the transcriptome. Upstream and downstream coding sequences are co-regulated by translational coupling. Coupling between translation and mRNA stability exists. And some genes even utilize multiple in-frame start codons to express multiple protein isoforms. Again, the authors' measurements can't distinguish the regulated ways in which the ribosome bound to their mRNAs. Simply measuring "the translation rates" of these coding sequences doesn't necessarily improve our ability to understand these mechanisms, certainly not in a way that would enable us to predict genetic part or genetic system function as the authors describe in their introduction. This is another reason why the authors need to tone down their perspective that NGS approaches will be the (one true) solution to the stated challenges (while dismissing other approaches).

It was not our aim to suggest that sequencing was the only true way of measuring transcriptional and translational processes. To address the Reviewer's concerns, we have significantly edited the Introduction to tone down our presentation of the benefits of sequencing. However, do believe that it is essential for a reader understand the limitations of widely used fluorescence-based methods, especially in regard to making high-throughput genome-wide measurements (a feature necessary for analysing genomes or large genetic circuits).

4. The authors mention in their results that "For example, the most metabolically efficient way to strongly express a protein of interest in bacteria is by producing high numbers of transcripts (e.g. with high transcription initiation rate and high stability) with a relatively weak RBS (e.g. low translation initiation rate)." This is not true. Producing large numbers of mRNA transcripts is a metabolic cost that is not necessary if the mRNA can support both a high translation initiation and elongation rate without compromising protein folding. However, if the mRNA can't be recoded to support high translation elongation rates, or if the protein is prone to misfolding, due to ribosome-ribosome interactions, then the authors' strategy is a good one. But there are no generalities here. Only a minority of *E. coli* proteins undergo co-translational protein folding, and most of them involve membrane interactions (i.e., not the transcription factors or enzymes commonly found in engineered genetic systems).

In light of the Reviewer's comments we have weakened our statement to say: “For example, a metabolically efficient way to strongly express a protein of interest in bacteria is by producing high numbers of transcripts (e.g. with high transcription initiation rate and high stability) with a relatively weak RBS (e.g. low translation initiation rate).” That said, we respectfully disagree with the Reviewer that only a small fraction of the bacterial proteome folds co-translationally. A cumulative knowledge over two decades has generated a comprehensive and quantitative picture of protein folding in the *E. coli* cell. In total, only 20-35% of all proteins fold truly post-translationally in either a DnaK/J (10-20%) or GroEL/ES (10-15%) dependent fashion. In contrast, 65-80% fold co-translationally (see review by Hartl, F.-U. and Hayer-Hartl M. *Science*, 2002). Approximately 40% are membrane proteins, whose folding as suggested by work from Art Johnson's, Roland Beckmann's and Gunnar von Heijne's laboratories can also occur co-translationally, even in the ribosomal tunnel. Co-translational protein folding and interactions with co-translationally binding chaperones is indeed facilitated by optimal elongation rates, which might not be necessarily the highest ones across a transcript. Moreover, to facilitate folding and interactions with auxiliary factors ribosomes elongate mRNA with a non-uniform speed littered with transient pauses over the sequence. Thus, high initiation rate would create collisions among the transiently paused preceding ribosomes; collisions are unproductive events as they cause ribosomal drop-off and premature

termination of synthesis which in turn is highly resourcefully and energetically disadvantageous.

Reviewer #3:

In my opinion, the authors have done a very reasonable job of addressing my concerns and have significantly improved the discussion of their results and methodology. The incorporation of gene-specific degradation rates is major improvement.

We thank the Reviewer for their careful consideration of our work and valuable comments, which we also believe have greatly improved the work.

Accepted

15th April 2019

Thank you again for sending us your revised manuscript. We are now satisfied with the modifications made and I am pleased to inform you that your paper has been accepted for publication.

Corresponding Author Name: Thomas E. Gorochowski

Manuscript Number: MSB-18-8719